# A cyclin-dependent kinase-mediated phosphorylation switch of disordered protein condensation

Juan Manuel Valverde [1,2,11], Geronimo Dubra [3,4,11], Michael Phillips [5], Austin Haider[6], Carlos Elena-Real[7], Aurélie Fournet[7], Emile Alghoul [8], Dhanvantri Chahar[3,4], Nuria Andrés-Sanchez[3,4], Matteo Paloni [5], Pau Bernadó [7], Guido van Mierlo [9], Michiel Vermeulen [9], Henk van den Toorn [1,2], Albert J. R. Heck [1,2], Angelos Constantinou [8], Alessandro Barducci [7], Kingshuk Ghosh[5,6], Nathalie Sibille [7], Puck Knipscheer [10], Liliana Krasinska [3,4,12], Daniel Fisher [3,4,12] ✉ & Maarten Altelaar [1,2,12] ✉

Cell cycle transitions result from global changes in protein phosphorylation states triggered by cyclin-dependent kinases (CDKs). To understand how this complexity produces an ordered and rapid cellular reorganisation, we generated a high-resolution map of changing phosphosites throughout unperturbed early cell cycles in single *Xenopus* embryos, derived the emergent principles through systems biology analysis, and tested them by biophysical modelling and biochemical experiments. We found that most dynamic phosphosites share two key characteristics: they occur on highly disordered proteins that localise to membraneless organelles, and are CDK targets. Furthermore, CDK-mediated multisite phosphorylation can switch homotypic interactions of such proteins between favourable and inhibitory modes for biomolecular condensate formation. These results provide insight into the molecular mechanisms and kinetics of mitotic cellular reorganisation.

The cell cycle is driven by CDKs, which are essential to promote entry into S-phase and mitosis. Two general strategies have been used to understand how CDK-dependent phosphorylation brings about these transitions[1]. First, top-down screens have revealed key system components. Hundreds of CDK substrates[2–7] and cell cycle-regulated proteins[8] and thousands of mitotic phosphorylations[9] have been

identified in this manner. Yet determining their roles has lagged behind; for example, painstaking genetic analysis in yeast models was required to reveal the requirement for CDK-mediated phosphorylation of two key substrates to allow DNA replication[10,11]. Second, bottom-up molecular analysis of the structural effects of individual phosphorylations on single proteins provides mechanistic insight into

¹Biomolecular Mass Spectrometry and Proteomics, Bijvoet Center for Biomolecular Research and Utrecht Institute for Pharmaceutical Sciences, University of Utrecht, Utrecht, 3584 CH Utrecht, Netherlands. ²Netherlands Proteomics Center, Padualaan 8, 3584 CH Utrecht, Netherlands. ³IGMM, CNRS, University of Montpellier, INSERM, Montpellier, France. ⁴Equipe Labellisée LIGUE 2018, Ligue Nationale Contre le Cancer, Paris, France. ⁵Department of Physics and Astronomy, University of Denver, Denver, Co 80208, USA. ⁶Department of Molecular and Cellular Biophysics, University of Denver, 80208 Denver, Co, USA. ⁷CBS, CNRS, University of Montpellier, INSERM, Montpellier, France. ⁸IGH, CNRS, University of Montpellier, Montpellier, France. ⁹Department of Molecular Biology, Faculty of Science, Radboud Institute for Molecular Life Sciences, Oncode Institute, Radboud University Nijmegen, Nijmegen 6525 GA, The Netherlands. ¹⁰Oncode Institute, Hubrecht Institute–KNAW and University Medical Center, Utrecht 3584 CT, Netherlands. ¹¹These authors contributed equally: Juan Manuel Valverde, Geronimo Dubra. ¹²These authors jointly supervised this work: Liliana Krasinska, Daniel Fisher, Maarten Altelaar. ✉e-mail: daniel.fisher@igmm.cnrs.fr; m.altelaar@uu.nl

regulation of their activity[12]. Both of these approaches are rendered more difficult by the fact that CDKs often phosphorylate multiple sites, whose combined effects may result in a phenotype; for example, multisite phosphorylation of the CDK inhibitor Sic1 in budding yeast is required to prevent its degradation[13]. As such, it has proven challenging to use studies performed from these different perspectives to understand global cellular behaviour.

Different models of CDK-mediated phosphorylation have been proposed. Specific interactions between distinct CDK-cyclin complexes and sequence motifs encoded in substrates might result in highly ordered phosphorylation[14], yet such complex mechanisms may not be essential since the cell cycle can be driven by single CDK1-cyclin complexes[15,16]. Furthermore, theoretical modelling and biochemical analysis have suggested that the mitotic control network can trigger switch-like activation of CDK1[17], yet approaches using fluorescent biosensors imply that mitotic CDK1 activation is rather progressive[18]. Along with the CDK-mediated ordered phosphorylation model, this suggests that mitotic phosphorylation states should change progressively in vivo, yet mitotic reorganisation is rapid and abrupt. Thus, understanding cell cycle transitions requires a description of the dynamics of global mitotic phosphorylation, which is largely unknown, as well as an investigation of the biochemical effects of phosphorylations, most of which remain uncharacterised. We reasoned that this would require a multidisciplinary quantitative approach involving cell biology, biochemistry, bioinformatics and biophysics. We aimed to generate a quantitative time-resolved map of in vivo cell cycle phosphorylation, extract global principles of phosphorylation dynamics, perform comparative analysis to assess conservation of these principles across species, and analyse effects of phosphorylation by modelling, biophysical approaches and biochemical experiments.

## Results

### A high-resolution map of in vivo cell cycle phosphorylation

Dynamic phosphorylation states cannot be determined from cell populations[19], while single-cell proteomics studies[20,21] currently have insufficient sensitivity and reproducibility for low stoichiometry and dynamic targets. We wanted to identify and quantify cell cycle-regulated phosphosites in a system devoid of artifacts arising from cell synchronisation[22,23], and with temporal resolution that alternative approaches, like centrifugal elutriation[24] or FACS[25] cannot provide. We thus took advantage of the naturally synchronous early cell cycles of *Xenopus laevis* embryos[26,27]. We performed quantitative phosphoproteomics in vivo using a sensitive phosphopeptide enrichment strategy[28]. We collected single embryos at 15-min intervals while recording visual cues of cell divisions, and phosphopeptides from each embryo were purified and analysed by mass spectrometry (Fig. 1a). These individual embryo phosphorylation states strongly correlated (Supplementary Fig. 1a). We identified 4583 high-confidence phosphosites mapping to 1843 proteins (Supplementary Fig. 1b; Supplementary Data 1), most being phosphoserines (Supplementary Fig. 1c). Our approach thus allowed us to generate a dynamic map of protein phosphorylation from an unfertilised egg to a 16-cell embryo.

We focused on 1032 sites on 646 proteins whose phosphorylation state changed over time (hereafter denoted "dynamic phosphosites"). Gene ontology (GO) and network analysis of proteins harbouring these sites revealed high functional association and interconnectivity between groups of proteins involved in RNA binding and the nuclear pore complex (NPC), DNA replication and chromatin remodelling, and microtubule regulation (Fig. 1b). Hierarchical clustering uncovered four groups with distinct dynamic behaviour (Fig. 1c; Supplementary Data 1). The levels of clusters A and B phosphosites were highest in eggs and post-fertilisation, and decreased during the first round of DNA replication. GO analysis for group A highlighted proteins involved in RNA regulation and nuclear organisation, including the NPC and nuclear transport, chromosomal structure and segregation (Supplementary

Fig. 1d), as also observed in a recent study on meiosis exit[29]. Cluster B phosphosites were enriched in regulators of RNA biosynthesis and stability, translation, actin, DNA replication and repair (Supplementary Fig. 1d). Dephosphorylation of these sites, which coincide with the meiosis-zygote transition, may prepare the embryo for upcoming cell divisions[30]. Cluster C phosphosites progressively increased after meiotic exit, while cluster D phosphosites had a clear oscillating signature with upregulation preceding each cell division. GO analysis of cluster C showed dominance of interphase cell cycle processes including DNA replication, RNA-related processes and chromosome organisation (Supplementary Fig. 1d). Several sites were from monophosphorylated peptides, while the multiphosphorylated forms were found in clusters A or D (Supplementary Fig. 1e), further indicating that cluster C sites are from interphase. Cluster D shows coordinated phosphorylation of multiple members of protein complexes involved in distinct processes such as nuclear transport, RNA processing, chromatin remodelling and DNA replication, suggesting a common mechanism of regulation (Supplementary Fig. 1f). Importantly, phosphoproteome changes were not simply a reflection of changes in abundance of the corresponding proteins (Supplementary Fig. 2), which are generally negligible during *Xenopus* early development[31]. Together, these results suggest that multisite phosphorylation, rather than progressive phosphorylation, might constitute the mitotic trigger.

To investigate this idea further, we assigned in vivo embryo phosphosites to different cell cycle stages by comparing with phosphorylation patterns of replicating or mitotic egg extracts (Fig. 1d). Replication was initiated by adding purified sperm chromatin to interphase egg extracts and quantified over time by measuring radioactive nucleotide incorporation (Fig. 1e, top), while mitosis was triggered by adding recombinant cyclin B and verified microscopically. To compare meiotic exit with mitosis, we also used egg extracts arrested at meiotic metaphase II (Cytostatic Factor, CSF-arrested). Overall, we identified 6937 phosphosites, which included 71% of the sites identified in vivo (Fig. 1f, Supplementary Data 1). 1728 sites varied between S and M-phase, including 693 sites upregulated in S-phase and 1035 in mitosis (Fig. 1e, Supplementary Data 1). The S-phase-specific phosphosites detected in this dataset greatly increase the known repertoire of phosphorylation sites upregulated during DNA replication[9]. Although several DNA-replication factors, including MCM4 and RIF1, were already phosphorylated in S-phase, they displayed marked multisite phosphorylation in mitosis (Supplementary Fig. 3a). GO analysis of interphase and mitotic sites revealed processes enriched in in vivo cluster C and cluster D, respectively (Supplementary Fig. 3b).

We next analysed the cell cycle behaviour of the dynamic phosphosites that we identified in vivo (Supplementary Fig. 3c). Most embryo cluster A sites were upregulated in both CSF-arrested meiotic extracts and mitotic extracts, highlighting the global similarities of meiotic and mitotic M-phase, despite the additional activity of the Mos/MEK/MAP kinase pathway in meiosis. In contrast, 78% of embryo cluster B sites were not mapped in extracts, suggesting that these are specific to the developmental transition from meiosis to early embryogenesis. In agreement, cluster B sites that mapped to extracts were generally low in mitosis. Most sites that mapped from embryo cluster C were present only in S-phase, whereas a subfraction peaked in mitosis. As such, this analysis revealed that clusters B and C do not strictly identify cell cycle phases. In contrast, all mapped cluster D sites were absent in S-phase and high in M-phase in vitro, showing that single embryo data can successfully identify mitotic phosphorylations. Next, we assessed the behaviour of the 102 peptides detected in both singly and multi-phosphorylated forms in extracts, and found that they segregated into two distinct clusters corresponding to interphase and M-phase, with a strong enrichment for multisite phosphorylation in M-phase (odds ratio 4.6, adjusted $p$-value $< 10^{-6}$) (supplementary Fig. 3d). The same analysis of multisite phosphorylation on the 35 peptides detected in singly or multiply phosphorylated forms in

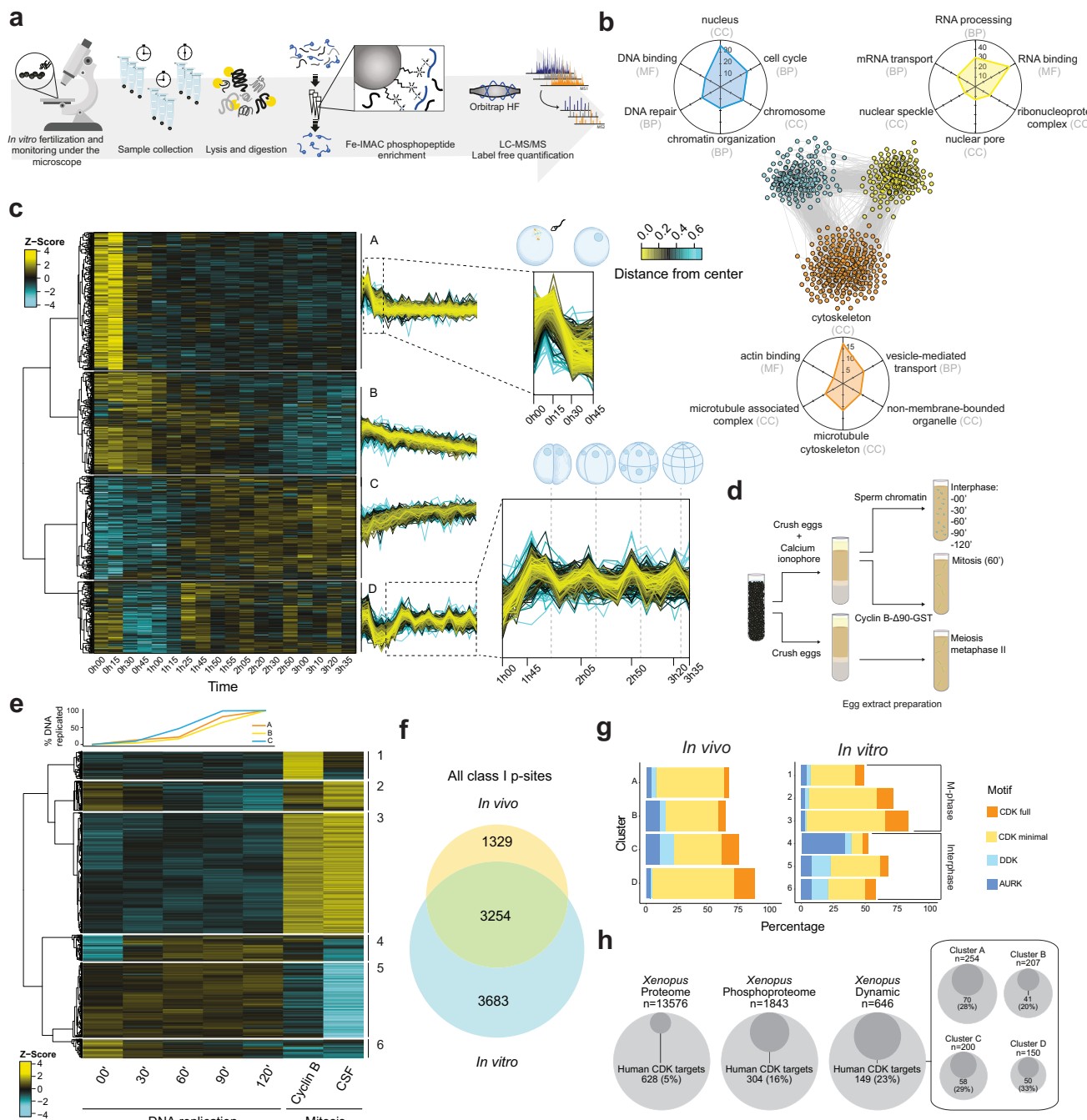

**Fig. 1 | The time-resolved phosphoproteome from a single-cell to a 16-cell embryo and its cell cycle assignment. a** Schematic representation of the workflow. Single *Xenopus* eggs and embryos were collected followed by cell lysis, protein digestion, phosphopeptide enrichment and high-resolution proteomics analysis. **b** STRING network of functionally associated proteins undergoing dynamic phosphorylation (each node represents a protein). Vicinity clustering reveals three main groups (yellow, blue and orange) with a high degree of association. Radar plots show the corresponding GO terms (Benjamini–Hochberg adjusted *p* value < 0.05) for each group (axes show −Log$_{10}$(adj *p* value) for each GO term). **c** Hierarchical clustering of significantly changing phosphosites (ANOVA, Benjamini–Hochberg correction, FDR 0.05), reveals 4 clusters with distinct regulation (A–D). Dashed boxes in clusters A and D are zoomed-in to highlight dynamic phosphorylation patterns (dashed lines depict the time points of cell division). Time point 0h00 corresponds to the unfertilised egg. **d** Scheme of the experiment in the *Xenopus* egg extract. **e** Top: quantification of DNA replication in each biological replicate. Below: Hierarchical clustering of dynamic phosphosites (ANOVA, Benjamini–Hochberg correction, FDR 0.05) reveals differential regulation of phosphosites during S-phase and mitosis. **f** Overlap between in vivo (embryo) and in vitro (egg extract) phosphoproteomics. **g** Proportion of phosphosites according to their potential upstream kinase for each cluster in the in vivo (left) and in vitro (right) experiments; DDK, Dbf4-dependent kinase, AURK, Aurora kinase. **h** Circle plots presenting enrichment of homologues of human CDK substrates among *Xenopus* phosphoproteins detected in vivo and those with dynamic phosphosites. Source data are provided as a Source data file.

embryos revealed four clusters, which correspond to clusters A–D (Supplementary Fig. 3e). Despite the small sample, multisite phosphorylation was highly enriched in meiotic metaphase cluster A (odds ratio 5.6, adjusted *p*-value 0.011), while two peptides were present as singly phosphorylated in interphase cluster C and doubly phosphorylated in mitotic cluster D. Overall, these data show that diverse biological processes may share a common regulatory mechanism of multisite phosphorylation during cell cycle progression.

## Predominance of CDK targets

We further investigated our detected phosphosites to identify putative kinases responsible for their dynamic phosphorylation. Analysis of kinase consensus motifs showed that proline-directed (S/T-P) sites, which conform to the minimal consensus for CDKs, constitute 51% of all detected phosphosites in vivo and 60% of dynamic sites (Supplementary Fig. 4a). Around 10% of all phosphosites matched the full CDK1-family consensus motif (S/TPxK/R). Replicating and mitotic extracts displayed a similar trend (Supplementary Fig. 4a). Putative CDK targets dominated all clusters, with 80% of sites in cluster D in vivo and mitotic clusters in vitro conforming to the minimal CDK motif (Fig. 1g, Supplementary Fig. 4b, c). Consensus sites of other kinases such as Aurora, Polo-like kinase (PLK), DBF4-dependent kinase (DDK) and Casein kinase I and II were present to a lesser extent (Supplementary Fig. 4b, d). The fraction of S/T-P sites that correspond to the full CDK consensus site motif is lower in clusters A (6%) and B (11%) than in clusters C and D (26 and 21%, respectively). Many of the cluster B S/T-P sites are likely mediated by MAP kinases, which are proline-directed kinases that are active in meiotic M-phase but are inactivated during early embryonic cell cycles[32]. In agreement with this idea, the MAP kinase consensus motif (GPLSP)[33] is clearly detectable in cluster B (Supplementary Fig. 4c). In contrast, around half of cluster A sites mapped to extracts are present in mitosis; this cluster has an unusual S/T-P motif that may correspond to a new class of CDK sites phosphorylated with distinct kinetics from cluster D sites.

Few direct CDK substrates have been characterised in *Xenopus*, but we surmised that CDK sites are likely conserved between vertebrates. Thus, to further analyse the proportion of dynamic sites dependent on CDKs, we manually curated a set of 654 human CDK1-subfamily targets (Supplementary Data 2; see 'Methods' for sources). 304 of these have *Xenopus* homologues among the 1843 phosphoproteins we detected, and 149 were present among the 646 proteins with dynamic phosphosites in *Xenopus* embryos (Fig. 1h). Thus, the predominance of CDK motifs among dynamic phosphosites reflects a high proportion of bona fide CDK substrates. This is a conservative estimate, since we only considered proline-directed sites as CDK motifs, although we found that 10–20% of human and yeast CDK substrates (Supplementary Data 2; see 'Methods' for sources) were non-proline-directed (Supplementary Fig. 4e), confirming a recent finding[34]. These data reinforce the dominant role of CDKs in cell cycle-regulated phosphorylation.

## Mechanisms generating synchronous mitotic phosphorylation in vivo

Since contrasting models of CDK activity predict either switch-like or progressive dynamics of substrate phosphorylation, we determined the in vivo dynamics of mitotic phosphorylation of individual phosphosites at extremely high-time resolution. We focused on 64 cluster D sites from diverse proteins, 60 of which conform to CDK minimal consensus motifs. We analysed these sites in single embryos every 180-s using quantitative targeted phosphoproteomics[35–37] by parallel reaction monitoring[38] (Fig. 2a), and thus obtained a highly resolved quantitative description of mitotic phosphorylation of different protein complexes in vivo. This revealed an exceptionally synchronous transition from low to maximal phosphorylation of all phosphosites preceding each cell division (Fig. 2b, c). The latter is as close to switch-like behaviour (an all-or-nothing response to a small change in regulator activity) as can be expected given the spatial metachronicity of mitotic entry in early *Xenopus* embryos[39,40]. Therefore, any differences in affinities of CDKs for substrates in vitro[14] do not translate into major timing differences in mitotic phosphorylation in vivo.

Highly synchronous mitotic phosphorylation of diverse proteins did not require oscillation of CDK1-Y15 inhibitory phosphorylation, which was downregulated over time (Fig. 2d), as previously reported[41], consistent with lack of oscillating phosphorylation of the CDK1-Y15-

regulatory enzymes, CDC25 and WEE1 (Fig. 2e). In contrast, oscillating phosphorylations on NIPA and the APC/C, which regulate mitotic cyclin accumulation, as well as Greatwall kinase, which activates the PP2A inhibitors Arpp19/ENSA, were apparent (Fig. 2f). This suggests that, in early embryos, control of mitotic cyclin levels and PP2A activity, and thus the overall CDK/phosphatase activity ratio[17], may be the key determinant of substrate phosphorylation timing and generate synchronous mitotic phosphorylation, whereas regulated CDK1-Y15 phosphorylation is not essential (Fig. 2g). This is consistent with the self-sufficiency of futile cycles in generating switch-like network output in the absence of allosteric regulation[42], and suggests that multiple layers of regulation may have evolved to ensure robustness of the dynamics.

## The cell cycle phosphoproteome is intrinsically disordered

We next investigated whether the diverse dynamic phosphoproteins that we identified share common structural features facilitating CDK-mediated phosphorylation, and if so, whether this is conserved across species. Phosphosites in general are often located in intrinsically disordered regions (IDRs) of proteins[43], which is also true for yeast and mouse CDK sites[6,7,44]. However, previous analyses did not take into account the enrichment of serine, threonine and proline in disordered regions, which is consistently predicted across the entire proteome of *Xenopus*, human and yeast (Supplementary Fig. 5a). Therefore, to date, it is not clear whether the presence of phosphorylatable amino acids, or specifically their phosphorylation, is enriched in disordered regions. We corrected for this compositional bias (see 'Methods'), and found that phosphorylatable residues in IDRs were more highly phosphorylated than those in ordered regions (Fig. 3a–c). This enrichment was increased for proteins with at least one site displaying dynamic phosphorylation; the same was true for human CDK substrates (Fig. 3b, c). To estimate the differential phosphorylation of disordered sites globally, we calculated the ratio of dynamically phosphorylated (*Xenopus*) or CDK-phosphorylated (yeast, human) to non-phosphorylated serine and threonine in both disordered and structured regions (Supplementary Fig. 5b; see 'Methods'). This showed that cell cycle and CDK-regulated phosphorylation is highly skewed towards disordered regions (Fig. 3d, Supplementary Fig. 5c). We then asked whether this is also true for substrates of other protein kinases, again taking into account compositional bias. We analysed the mitotic PLK and Aurora kinases (which control various aspects of mitotic chromosome and spindle dynamics), DYRK kinases, (which promote mitotic phosphorylation of several intrinsically disordered proteins [IDPs])[45], NEK kinases (which have roles in centrosome duplication and various stages of mitosis), and MAP kinases (which share the proline-directed S/T minimal consensus site with CDKs). For each of these, with the exception of NEK kinase targets, documented phosphosites were strongly enriched in IDRs (Supplementary Fig. 5c, d), supporting the idea that phosphorylation of residues in IDRs is a general cellular control mechanism and is not specific to CDKs.

Given the preponderance of CDK substrates among cell cycle-regulated phosphosites despite the evidence that many kinases may preferentially phosphorylate IDRs, we wondered whether CDK substrates might be more disordered than phosphoproteins in general. We therefore determined the percentage of disordered residues of proteins in our datasets, compared to the rest of their respective phosphoproteomes (Supplementary Data 3). This revealed that, on average, both *Xenopus* dynamic phosphoproteins and human and yeast CDK substrates contain approximately twice the proportion of disordered amino acids as all other phosphoproteins (Fig. 3e, Supplementary Fig. 5e), putting them among the top quartile of proteins with the most disorder in the proteome. Furthermore, targets of most cell cycle kinases except NEK are significantly more disordered than targets of MAP kinase (Fig. 3f), whose phosphosites are also proline-directed and preferentially located in IDRs (Supplementary Fig. 5d).

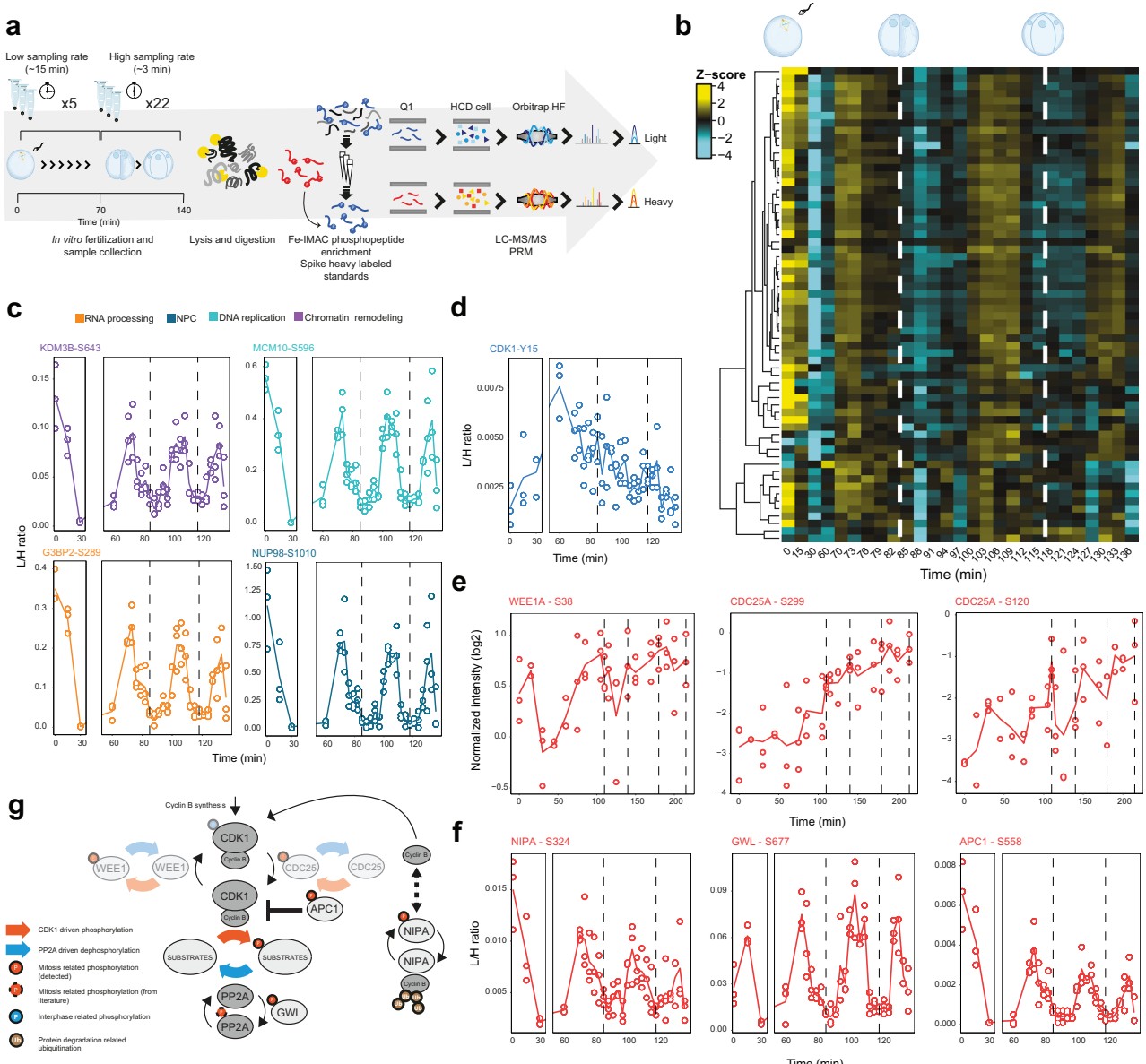

**Fig. 2 | Switch-like mitotic phosphorylation in vivo. a** Schematic representation of the workflow for high-time resolution analysis of mitotic phosphosites. Samples were collected over two cell divisions and enriched phosphopeptides were subjected to targeted proteomics analysis. **b** Heat map shows a highly synchronous wave of phosphorylation preceding each of the two cell divisions. Dashed lines depict times when cell divisions were recorded. **c** Single phosphosite plots from selected proteins related to different biological processes. Each dot represents a biological replicate ($n = 3$). Dashed lines depict times when cell divisions were recorded. **d** Single phosphosite plot of CDK1 inhibitory phosphorylation (Y15). **e**–**g** Phosphorylation dynamics of the CDK1-oscillator network. Single phosphosite plots of CDK1 regulators measured by shotgun (**e**) or targeted (**f**) phosphoproteomics. **g** CDK1-oscillator network: our data suggest that control of cyclin levels via positive (e.g. NIPA ubiquitin ligase) and negative (e.g. APC, anaphase-promoting complex) feedback loops, accompanied by protein phosphatase 2A (PP2A) inactivation via Greatwall kinase (GWL), can generate oscillation of CDK1 activity during early cell divisions. CDK1-Y15 regulation via feedback loops consisting of CDC25 phosphatase and Wee1-like protein kinase 1-A (WEE1A; greyed out) seems to be less important for switch-like mitotic phosphorylation after the first cell division. Source data are provided as a Source data file.

This suggests that several cell cycle kinases have evolved to phosphorylate some of the most disordered proteins in the proteome.

**Enrichment of MLO components among CDK substrates**
We surmised that the critical importance of intrinsic disorder might underlie a common mechanism of phosphoregulation of diverse proteins during the cell cycle. We noted that IDPs are key components of membrane-less organelles (MLO), many of which (*e.g.* Cajal bodies, nucleoli, nuclear pore complexes, splicing speckles) show cell cycle-regulated assembly or disassembly and are thought to arise by phase separation (PS)[46] that can be controlled by phosphorylation[45,47–49]. We

thus hypothesised that CDKs might regulate PS during the cell cycle. We analysed available data on cellular localisation for each of our curated human CDK substrates, and found that 257 (39.2%) are present in MLOs (Fig. 4a). We then manually curated an MLO proteome from human proteomics studies and asked whether proteins undergoing cell cycle-regulated phosphorylation are enriched in these compartments (Supplementary Data 4; see 'Methods' for sources). Indeed, homologues of 204 dynamic *Xenopus* phosphoproteins (31.6%) localise to MLOs, as do 73 of the 149 proteins (50%) that show dynamic phosphorylation in *Xenopus* and are CDK substrates in human (Fig. 4a). Of the latter, we studied the location of phosphosites on key IDPs of

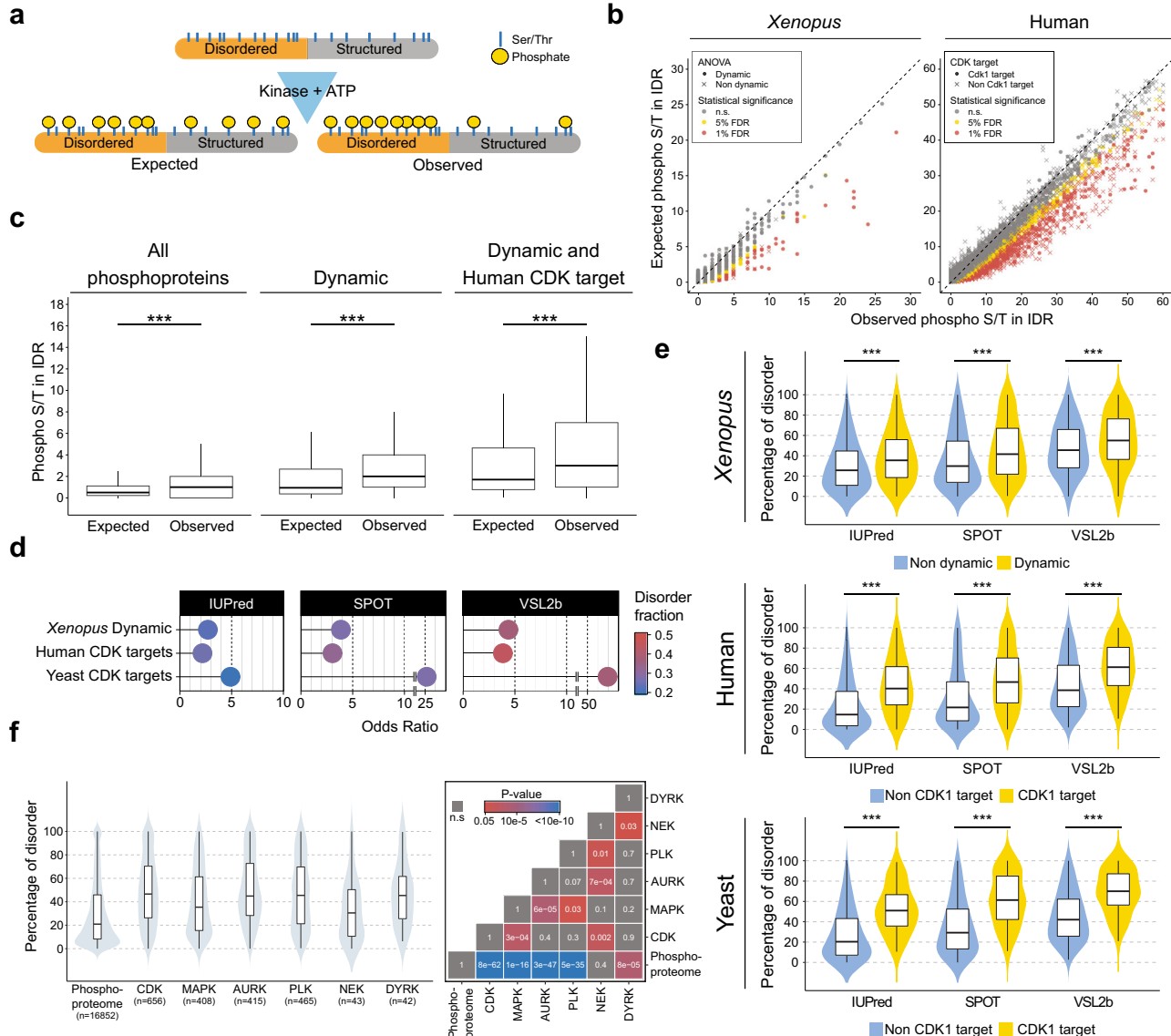

**Fig. 3 | The cell cycle phosphoproteome is characterised by intrinsic disorder and MLO components. a** Scheme illustrating hypothetical enrichment of phosphorylation in disordered regions when taking into account amino acid compositional bias. **b** Scatter plot of expected vs observed phosphorylated Ser/Thr for each protein of human and *Xenopus* phosphoprotein datasets. FDR thresholds of 5% and 1% are marked in yellow and red respectively. Circles: proteins with at least one dynamic phosphorylation in *Xenopus,* or human CDK1 subfamily substrates, respectively. **c** Boxplots showing expected vs observed phosphorylated Ser/Thr in IDRs (IUpred) among all phosphoproteins detected (left), phosphoproteins with at least one dynamic phosphosite (middle), and dynamic phosphoproteins also detected as CDK1 subfamily targets in humans (right). Distributions were compared with the paired Wilcoxon signed-rank two-sided test. All phosphopoteins (1843), *p* < 2.2e−16; Dynamic (646), *p* < 2.2e−16; Dynamic and human CDK target (147), *p* = 1.0e−14. Boxplots centre, median; lower and upper edges, 25% and 75% quartiles, respectively. Whiskers, data with the largest or smallest values not further than 1.5* interquartile range (IQR) from the upper or lower box limits, respectively. Beyond these values, data were not plotted, for clarity, but were included in the statistical analysis. **d** Plots showing the common Odds Ratio of Ser/Thr phosphorylation in structured and disordered regions calculated with the Fisher's test

(see Supplementary Fig. 5b, c for the statistical analysis scheme and exact p values). For all organisms, the disordered regions were calculated with three different disorder predictors. The disordered fraction is presented in a colour scale. **e** Violin plots of the distribution of disordered residues per protein for CDK targets vs the rest of the phosphoproteome for human and yeast, and dynamic phosphoproteins vs the rest of the phosphoproteome for *Xenopus*. Intrinsic disorder was calculated with three different predictors (IUPred, SPOT, and VSL2b). Statistical significance was evaluated with the Wilcoxon−Mann−Whitney signed-rank two-sided test: *Xenopus* (CDK targets, 646, Non-CDK targets, 1198): IUpred *p* = 2.36e−13, SPOT *p* = 4.83e−13, VSL2b *p* = 5.78e−10; Human (CDK targets, 16167, Non-CDK targets, 2153): IUpred *p* < 2.2e−16, SPOT *p* < 2.2e−16, VSL2b *p* < 2.2e−16; Yeast (CDK targets, 100, Non-CDK targets, 2153): IUpred *p* < 2.2e−16, SPOT *p* < 2.2e−16, VSL2b *p* < 2.2e−16. Boxplot parameters as in (**c**). **f** Violin plot (left) showing the distribution of disordered residues per protein (as estimated with SPOT) for CDK, MAPK, Aurora, PLK, NEK and DYRK kinase targets vs the rest of the phosphoproteome for human targets. Statistical significance was assessed by Kruskal−Wallis ANOVA, and pairwise comparisons were performed with Dunn's two-sided post hoc tests. The adjusted *p*-values (Benjamini−Hochberg) are shown in a tile plot (right). Boxplot parameters as in c. Source data are provided as a Source data file.

different MLOs, including coilin (Cajal bodies), nucleophosmin, nucleolin and Ki-67 (nucleoli), 53BP1 (53BP1 bodies), nucleoporins (NPC) and PML (PML bodies). As we anticipated, the vast majority of both proline-directed phosphosites and confirmed CDK sites on these proteins were located in predicted IDRs (Fig. 4b). We next investigated

whether dynamically phosphorylated IDPs have properties characteristic of phase separation. We first applied a machine learning classifier[50] to predict whether cell cycle-regulated phosphoproteins show higher propensity to phase separate (PSAP score). Indeed, we observed a sharp increase in the PSAP score, from the proteome to the

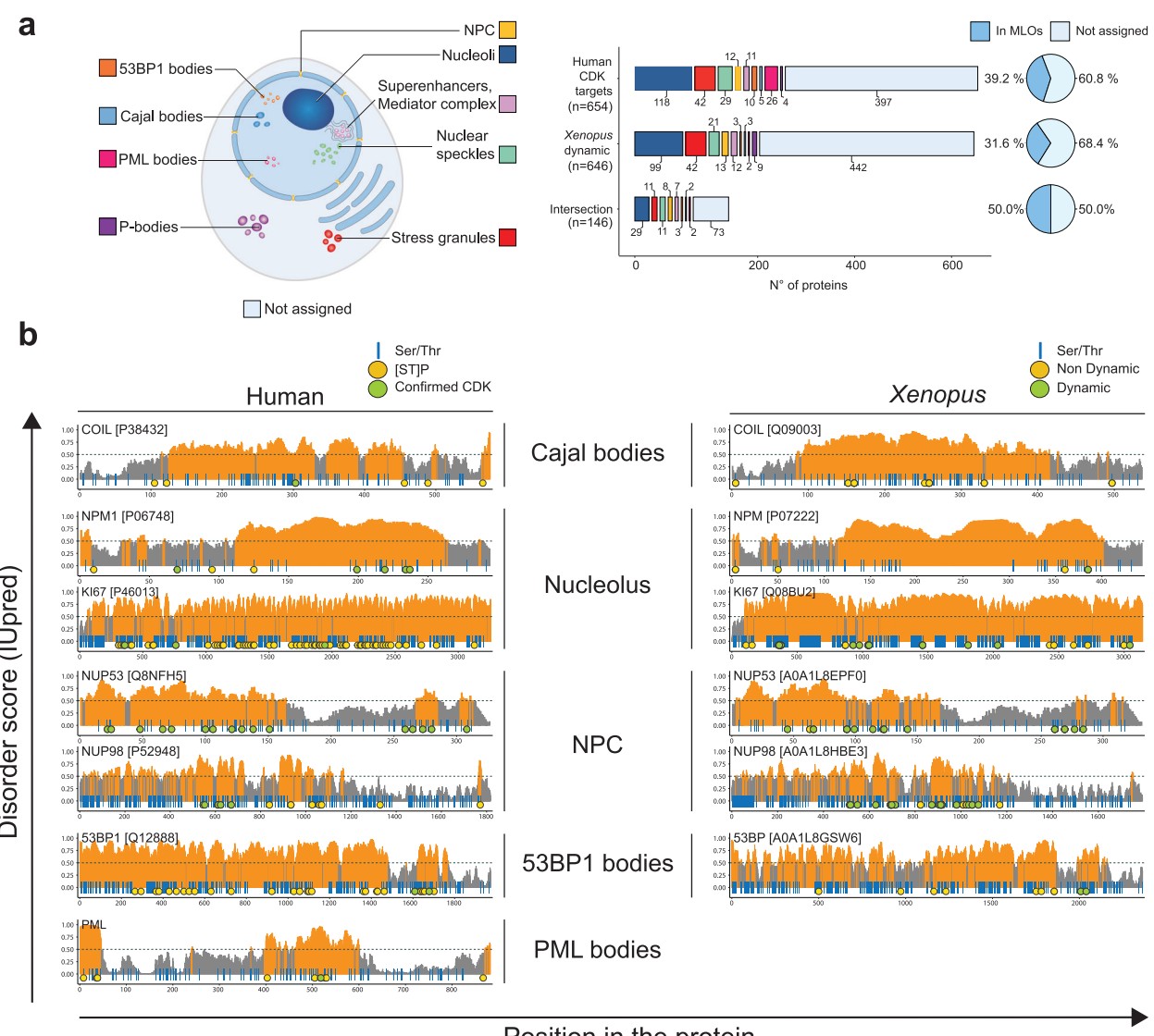

**Fig. 4 | Cell cycle-regulated phosphorylation of key MLO proteins. a** Human CDK1 subfamily targets, *Xenopus* dynamic phosphoproteins, and the intersection of both sets, that are present in our manually curated proteome of membraneless organelles (MLOs; NPC, nuclear pore complex; PML, promyelocytic leukemia protein). **b** Diagrams of IUPred scores over the length of human CDK targets identified as primary components of MLOs in different studies, and their *Xenopus* homologues in this study. Regions with scores >0.5 (orange) are considered to be disordered, and <0.5 (grey) structured. Blue vertical lines indicate Ser and Thr residues; yellow circles, known Ser/Thr-Pro phosphosites (human) and non-dynamic phosphosites (*Xenopus*); green circles, confirmed CDK1 subfamily phosphorylations (human) and dynamic phosphorylations (*Xenopus*), from both embryos and egg extracts. Source data are provided as a Source data file.

phosphoproteome, and a further increase for dynamic phosphoproteins, with the highest score for mitotic cluster D (Supplementary Fig. 6). Similarly, the PSAP score is far higher amongst targets of most cell cycle kinases (CDK, Aurora, PLK, but not NEK) and DYRK kinases than the overall phosphoproteome, but less so for MAP kinase substrates.

## CDKs regulate IDR phase separation

Both stochastic and specific interactions between IDPs contribute to PS and MLO assembly[46,51,52], and protein phosphorylation can promote or inhibit PS, depending on the protein sequence context[49,53]. Although most MLOs disassemble in mitosis, there is a notable exception: the perichromosomal layer (PCL), which has been hypothesised to form via PS. We reasoned that the degree of CDK-mediated multisite IDP phosphorylation might constitute a switch between PS promotion and inhibition. For example, the maximal CDK activity present at the onset

of mitosis might promote both disassembly of most MLOs and formation of the PCL. Studying the effects of CDK-mediated phosphorylation in a considerable number of diverse proteins is not feasible by biochemical approaches and is challenging even for molecular dynamics simulations. To overcome this obstacle, we employed analytical modelling. We took advantage of a newly developed mathematical theory called renormalised Gaussian random phase approximation (rG-RPA)[54], that combines traditional RPA theory with sequence-dependent single-chain theory using a renormalised Gaussian (rG) chain formulation. This theory provides a better account of conformational heterogeneity and density fluctuations, allows predictions of phase separation, and can be employed at a medium-throughput scale. We supported our findings by another recent theory that computes sequence charge decoration matrices (SCDM) to study conformational properties of a single IDP chain; this accounts for the effects of sequence-specific electrostatic interactions on chain

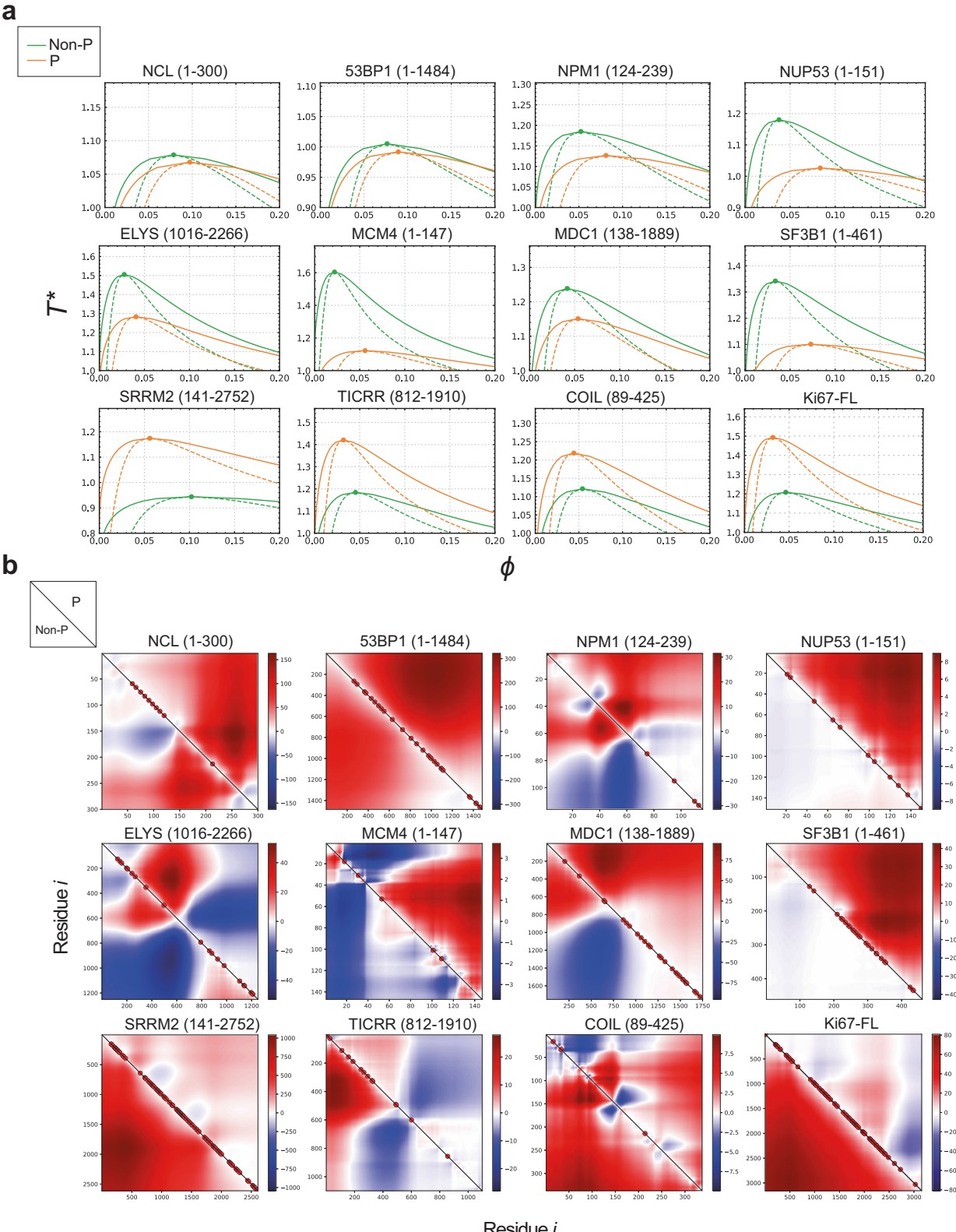

conformation, which dictate the distance between any two amino acid residues[55–57]. Since single chain properties can predict multi-chain physical behaviour, we expect SCDM to provide further insights into the propensity of phase separation. We chose 12 IDPs representing different biological processes and MLOs and containing multiple CDK phosphorylation sites; the choice of the disordered regions of these proteins was mostly based on AlphaFold2[58] (see 'Methods';

Supplementary Data 5). Phosphorylation of all described CDK-sites lowered critical temperature and phase separation propensity of 8 IDPs (NCL, NPM1, NUP53, ELYS, 53BP1, MCM4, MDC1, and SF3B1) (Fig. 5a). These trends are consistent with SCDM maps showing decreased self-association propensity (increased red regions, Fig. 5b). Conversely, for Ki-67, SRRM2, TICRR and coilin, CDK-mediated phosphorylation increased critical temperature and PS tendency (Fig. 5a),

**Fig. 5 | CDK-mediated phosphorylation regulates phase separation propensity of major MLO components. a** Temperature-density phase diagrams for the phosphorylated (P, in orange) and non-phosphorylated (Non-P, in green) forms of a selection of human CDK targets and major MLO components (the disordered regions analysed are indicated). Any point within the coexistence region (bounded by the solid line) will phase separate into a dilute and dense phase whose density is given by the values on the phase boundary. Points within the spinodal line (in dash) will spontaneously phase separate into dilute and dense phases without going through the process of nucleation. Circles denote critical temperature and density which is also the location where the coexistence and spinodal curves meet. For temperature above the critical value, there is no phase separation. **b** Sequence Charge Decoration Matrix (SCDM) maps for the proteins in (**a**), depicting the contribution of electrostatic interaction dictating the distance between two amino acid residues i and j (shown in x and y axes). The values of SCDM for different residue pairs (i,j) are shown using colour schemes with red and blue denoting positive (repulsive) and negative (attractive) values, respectively. The lower and upper triangles indicate SCDM map for the unphosphorylated (non-P) and phosphorylated (P) sequences, respectively. Confirmed and putative (Ser/Thr-Pro) CDK phosphorylation sites were taken into account for the analysis (in both **a** and **b**) and are indicated with red circles.

consistent with SCDM maps (increased blue or reduced red regions, Fig. 5b). Overall, these data are in agreement with our hypothesis that CDK-mediated phosphorylation is a key regulator of PS propensity of IDPs.

To analyse this in more detail, we focused on a model CDK substrate, Ki-67, an IDP that organises heterochromatin structure[59] and perichromosomal layer formation from nucleolar components in mitosis[60,61]. Ki-67 is highly phosphorylated in mitosis by CDKs, which regulates its perichromosomal localisation[62]. It contains a multivalent repeat domain of 122 amino acids, known as the Ki-67 repeat, each harbouring a highly conserved motif, the Ki-67 motif, currently of unknown function (Fig. 6a).

To evaluate the possibility that Ki-67 may show PS behaviour in living cells, we expressed a GFP-tagged full-length Ki-67 protein and compared its localisation pattern in cells with different expression levels (Fig. 6b). At lower levels, Ki-67 was detectable in the nucleolus, the site of endogenous Ki-67, but not in the nucleoplasm, while at higher levels, additional foci appeared outside the nucleolus. This suggests that Ki-67 is partitioned by PS into dense and dilute phases. At the highest levels, Ki-67 showed a virtually continuous condensed phase, reminiscent of spinodal decomposition[63] whereby the single phase spontaneously separates uniformly throughout the space into two phases, without the nucleation and growth which normally characterises phase separation. In these conditions, chromatin exhibited the same pattern (Fig. 6b), indicating that Ki-67 overexpression drives heterochromatinisation, consistent with our earlier findings[59]. We next examined whether Ki-67 mobility is consistent with liquid-like behaviour by Fluorescent Recovery After Photobleaching (FRAP) experiments. We investigated both the recovery of the bleached area of part of a Ki-67-positive compartment, and an adjacent unbleached area of the same compartment. Rapid recovery of the bleached area and preferential mixing of components from the unbleached to the bleached area would be indicative of a liquid-like phase-separated compartment[64]. We performed such FRAP assays on cells with different levels of Ki-67, in interphase, or in mitosis where Ki-67 localises to the PCL. At low expression levels, the recovery half-time was of about 14 s, while high expression levels showed a significantly slower recovery. This is consistent with the emerging idea that phase separation may be coupled with percolation, whereby a percolated network of molecules spanning the volume of condensates confers viscoelastic properties[65]. Interestingly, the recovery was extremely rapid in mitosis, of 8 s (Fig. 6c, d; Supplementary Movies 1–3). This behaviour indicates liquid-like mixing with an impermeable barrier of the phase-separated compartment[64], and suggests that the perichromosomal layer may be a highly liquid-like phase (Fig. 6c, d; Supplementary Movies 1–3). We further analysed the propensity of Ki-67 to undergo phase separation in cells by using the light-activated Cry2 "optodroplet" system[66] with full-length Ki-67 or a series of deletion mutants. Full-length Ki-67 localised to the nucleolus, as expected, but exposure to blue light caused rapid appearance of small round foci in the nucleoplasm, which was dependent on the level of induced Ki-67 expression, consistent with PS (Fig. 6e). These foci showed colocalisation with nucleolin and nucleophosmin (Fig. 6f), intrinsically disordered proteins and

interactors of Ki-67 involved in nucleolar organisation, indicating the existence of heterotypic interactions typical of phase-separated MLOs.

We next analysed both by modelling and experiments the consequences of phosphorylation for PS of Ki-67. As shown above, rG-RPA of full-length Ki-67 predicted that its complete phosphorylation should promote PS (Fig. 5a), in agreement with Coarse-grained (CG) molecular dynamics (MD) simulation, which showed increased compaction upon phosphorylation (Fig. 7a; Supplementary Movie 4; Supplementary Data 6), and with SCDM analysis, which indicated increased intra-chain interactions (Fig. 5b). We next tested effects of CDK-mediated phosphorylation on Ki-67 phase separation propensity using the optodroplet system in cells. To do this, we promoted the CDK-mediated phosphorylation state by treating cells with okadaic acid to inhibit PP2A, that reverses CDK-mediated phosphorylation, or inhibited CDKs with purvalanol A[17]. Even in the absence of blue light, treatment with okadaic acid led to formation of new Ki-67 foci within the nucleoplasm (Fig. 7b, c), which also incorporated nucleolin and nucleophosmin (Fig. 7d). Therefore, phosphorylation promotes PS of Ki-67 independently of oligomerisation of the Cry2 domain. Conversely, pan-CDK inhibition with purvalanol A prevented induction of foci upon illumination (Fig. 7b, c). These results are consistent with phosphorylation of full-length Ki-67 promoting PS, as predicted by SCDM, rG-RPA and MD. We observed similar behaviour for constructs lacking the C-terminal LR domain, that binds chromatin, or the N-terminal domain, which is required for the nucleolar localisation of Ki-67 (Supplementary Fig. 7a, b), indicating that PS of Ki-67 is an autonomous property of the protein and is not dependent on a specific localisation on chromatin or to the nucleolus.

A recent in vitro study using purified peptides corresponding to repeat domains showed that partial phosphorylation by CDKs of one of the Ki-67 repeats, or phospho-mimicking amino acid substitutions, promotes PS, possibly by generating charge blocks[67]. However, rG-RPA and SCDM analysis predicts that the effects of phosphorylation on PS propensity strongly depend on the sequence. To test this, we analysed each repeat individually by rG-RPA and SCDM, and indeed found that effects of phosphorylation were dependent on the particular repeat studied, as illustrated by the behaviour of repeats 1, 3 and 12. Full phosphorylation was predicted to enhance PS for repeat 1 but suppress it for repeat 3 (Supplementary Fig. 8a, b). Furthermore, we observed that phospho-mimicking substitutions, which each add one negative charge, do not recapitulate effects of phosphorylation, which adds two per site, and can even have opposite effects, as seen for repeat 12 (Supplementary Fig. 8a, b). These results indicate that effects of phosphorylation on PS depend on sequence context and phosphorylation stoichiometry.

We hypothesised that, for many IDPs, full site phosphorylation, which is characteristic of mitosis[9], might have different effects on PS from the lower levels of phosphorylation typical of interphase, thereby providing directionality to PS changes during the cell cycle and potentially creating a switch at mitosis. To test this hypothesis, we designed a synthetic IDP constituting a single "consensus" synthetic Ki-67 repeat derived from an alignment of all 16 Ki-67 repeats, in effect constituting a novel IDP and model CDK substrate with multiple

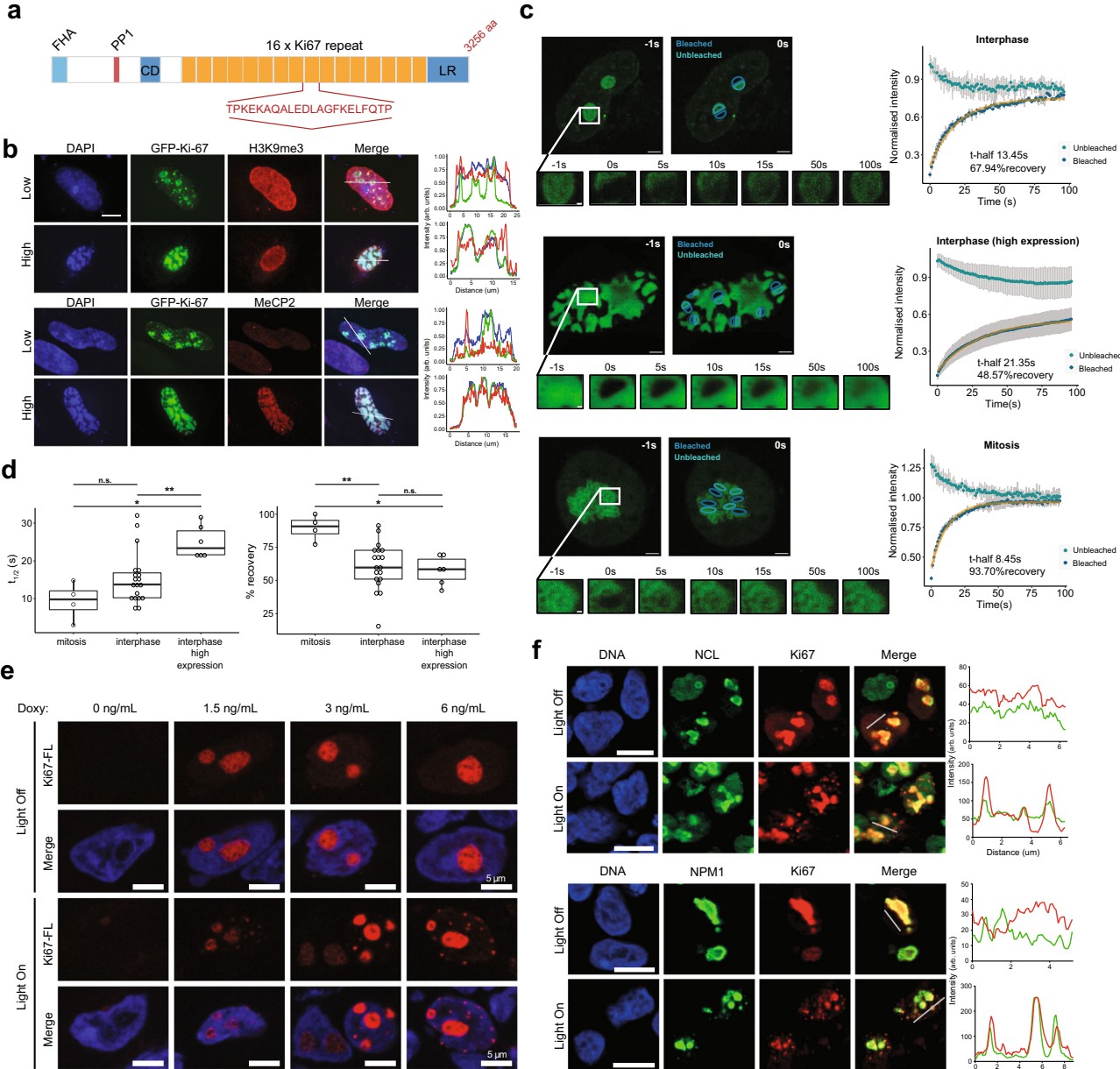

**Fig. 6 | The model CDK substrate Ki-67 forms biomolecular condensates in cells. a** Scheme of the human Ki-67 protein (FHA, forkhead-associated domain; PP1, PP1 phosphatase-binding domain; CD, conserved domain; LR, leucine arginine-rich domain; aa, amino acid). Highlighted, Ki-67 repeat consensus motif. **b** HeLa cells expressing full-length GFP-tagged Ki-67 at different levels show evidence for phase separation; at low levels Ki-67 is predominantly nucleolar but also forms foci in nucleoplasm that recruit heterochromatin, as indicated by H3K9me3 staining; at high levels Ki-67 partitions the entire nucleus into two phases, a Ki-67-dense phase that induces global heterochromatin formation marked by H3K9me3 and MeCP2. DNA was stained with DAPI. Colocalisation between DAPI (blue), GFP-Ki-67 (green), and either MeCP2 or H3K9me3 (red) is shown, right. Scale bar, 10 μm; $n = 2$ independent experiments. **c** FRAP of Ki-67 shows liquid-like behaviour. Left: representative images of cells expressing different levels of Ki-67 in interphase (top, middle) and in mitosis where Ki-67 localises to the perichromosomal layer (bottom) showing bleached regions and contiguous non-bleached regions; just before (left) and after (right) bleaching. Scale bars, 3 μm. Insets: images of Ki-67 fluorescence at different recovery times after bleaching. Scale bars, 0.5 μm. Right: average fluorescence intensity values over time for bleached (dark blue) and unbleached (teal blue) regions. Orange, non-linear regression fitting of the data; n = 2 independent experiments. Each line represents the dynamics of one cell, in which several regions were photobleached at time 0 and the intensity values over time were analysed, and a corresponding number of regions was not bleached but analysed (mitosis, 4

regions; interphase, 2 regions; interphase with high expression, 7 regions). The mean values of all replicates were plotted as normalised intensity vs time, grey error bars, standard deviation. **d** Left: boxplot of the average recovery half-time for each cell, grouped by category of Ki-67 expression: mitosis ($n = 4$), interphase medium-low expression ($n = 19$) and interphase high expression ($n = 6$). Right: boxplot of the average percentage of recovery for each cell, grouped by category of Ki-67 expression: mitosis ($n = 4$), interphase medium-low expression ($n = 19$) and interphase high expression ($n = 6$). **$p < 0.01$ (Wilcoxon test, two-sided); t-half: interphase vs mitosis, $p = 0.123$; interphase-HiExpr vs mitosis, $p = 0.014$; interphase-HiExpr vs interphase, $p = 0.008$; %recovery: interphase vs mitosis, $p = 0.005$; interphase-HiExpr vs mitosis, $p = 0.014$; interphase-HiExpr vs interphase, $p = 0.589$. **e** Optogenetic induction of Ki-67 biomolecular condensates. Representative fluorescent images of HEK-293 cells expressing opto-Ki-67 (FL) construct, induced by the indicated concentrations of doxycycline (Doxy), before (Light Off) and after (Light On) exposure to blue light. DNA was stained with Hoechst 33258; $n = 2$ independent experiments. **f** Left, representative fluorescent images of U2OS cells expressing opto-Ki-67 construct (full length protein) before (Light Off) and after (Light On) exposure to blue light. Additional staining for nucleolar proteins nucleolin (NCL, top) and nucleophosmin (NPM1, bottom) was performed and colocalisation with Ki-67 assessed (right; Ki-67, red; NCL or NPM1, green). DNA was stained with Hoechst 33258; scale bars, 10 μm; $n = 2$ independent experiments. Source data are provided as a Source data file.

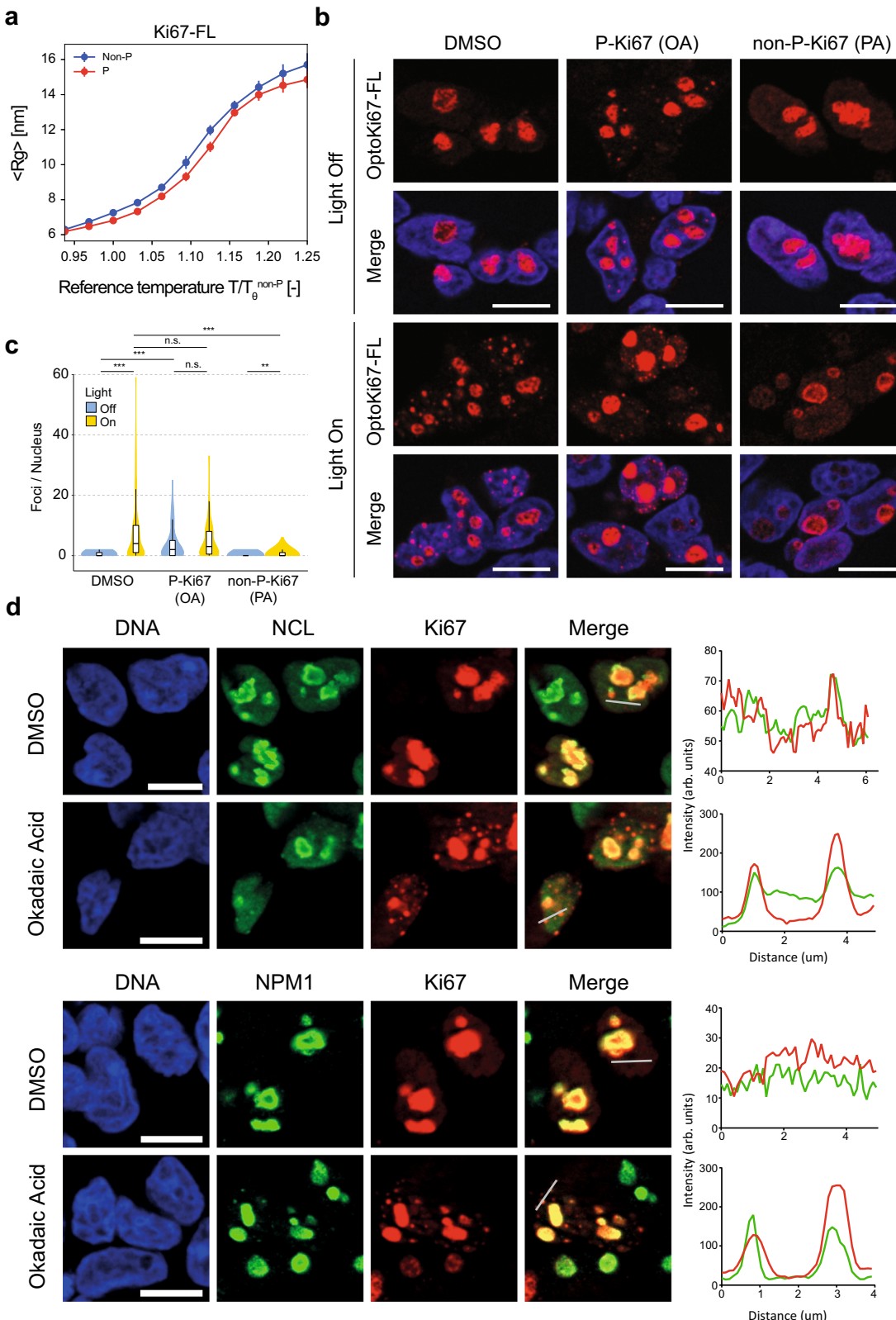

potential phosphosites (Supplementary Fig. 9a). We expected that this protein might show a different behaviour from that of the native full-length Ki-67 due to its single valency and distinct sequence context. Indeed, MD simulation (Supplementary Movies 5, 6) showed that the fully phosphorylated single synthetic Ki-67 repeat is less compact (Supplementary Fig. 9b) and has lower tendency to phase separate (Supplementary Fig. 9c). This behaviour contrasts with full-length

native Ki-67, where phosphorylation promotes PS. We purified the synthetic repeat as a GFP-tagged polypeptide and phosphorylated it in vitro with recombinant CDK complexes. Nuclear Magnetic Resonance spectroscopy showed a reduced amide proton spectral dispersion typical for an IDP and confirmed appearance of at least 7 phosphorylated residues when fully phosphorylated (Fig. 8a), while we mapped 11 phosphorylation sites by phosphoproteomics (Fig. 8b). We

**Fig. 7 | Phosphorylation promotes phase separation of Ki-67. a** Coarse-grained single-chain MD simulations for full chain Ki-67 showing dependency of the radius of gyration (Rg) on the simulation temperature. The reference temperature is the θ temperature of the non-phosphorylated molecule. Error bars correspond to the standard error of the mean obtained with block analysis by subdividing the trajectory into 10 non-overlapping blocks. **b** Representative fluorescent images of HEK-293 cells expressing opto-Ki-67 (FL) construct before (Light Off) and after (Light On) exposure to blue light. Cells were pretreated for 1 h with either vehicle (DMSO), 0.5 μM okadaic acid (OA), to inhibit protein phosphatase 2 A, or 5 μM purvalanol A (PA), to inhibit CDKs. DNA was stained with Hoechst 33258; scale bars, 10 μm; $n = 3$ independent experiments. **c** Violin plot presenting quantification of results from (**b**); the number of foci per nucleus was counted (≥100 nuclei per condition). Statistical significance was assessed by one-way ANOVA on ranks (Kruskal–Wallis test) and pairwise post hoc comparisons using the Mann–Whitney two-sided test. *P*-values were adjusted by the Benjamini–Hochberg method: DMSO Off vs DMSO On, $p = 2.1e{-}14$; OA On vs DMSO Off, $p = 0.315$; OA On vs DMSO Off,

$p = 8.6e{-}14$; OA Off vs DMSO On, $p = 0.023$; OA Off vs DMSO On, $p = 5.3e{-}11$; OA Off vs OA On, $p = 0.186$; PA On vs DMSO On, $p = 2.0e{-}15$; PA On vs DMSO Off, $p = 0.642$; PA On vs OA On, $p = 2.1e{-}14$; PA On vs OA Off, $p = 1.8e{-}11$; PA Off vs DMSO On, $p < 2e{-}16$; PA Off vs DMSO Off, $p = 7.0e{-}05$; PA Off vs OA On, $p < 2e{-}16$; OA Off vs OA Off, $p < 2e{-}16$; PA Off vs PA On, $p = 0.001$. Boxplots centre, median; lower and upper edges, 25% and 75% quartiles, respectively. Whiskers, data with the largest or smallest values not further than 1.5* interquartile range (IQR) from the upper or lower box limits, respectively. Beyond these values, data were not plotted, for clarity, but were included in the statistical analysis. **d** Phosphorylation-induced Ki-67 foci are biological condensates. Cells were treated for 1 h with either vehicle (DMSO) or 0.5 μM okadaic acid (OA), to inhibit protein phosphatase 2A. Additional staining for nucleolar proteins nucleolin (NCL) and nucleophosmin (NPM1) was performed and colocalisation with Ki-67 foci assessed (right; Ki-67, red; NCL or NPM1, green). DNA was stained with Hoechst 33258; scale bars, 10 μm; $n = 2$ independent experiments. Source data are provided as a Source data file.

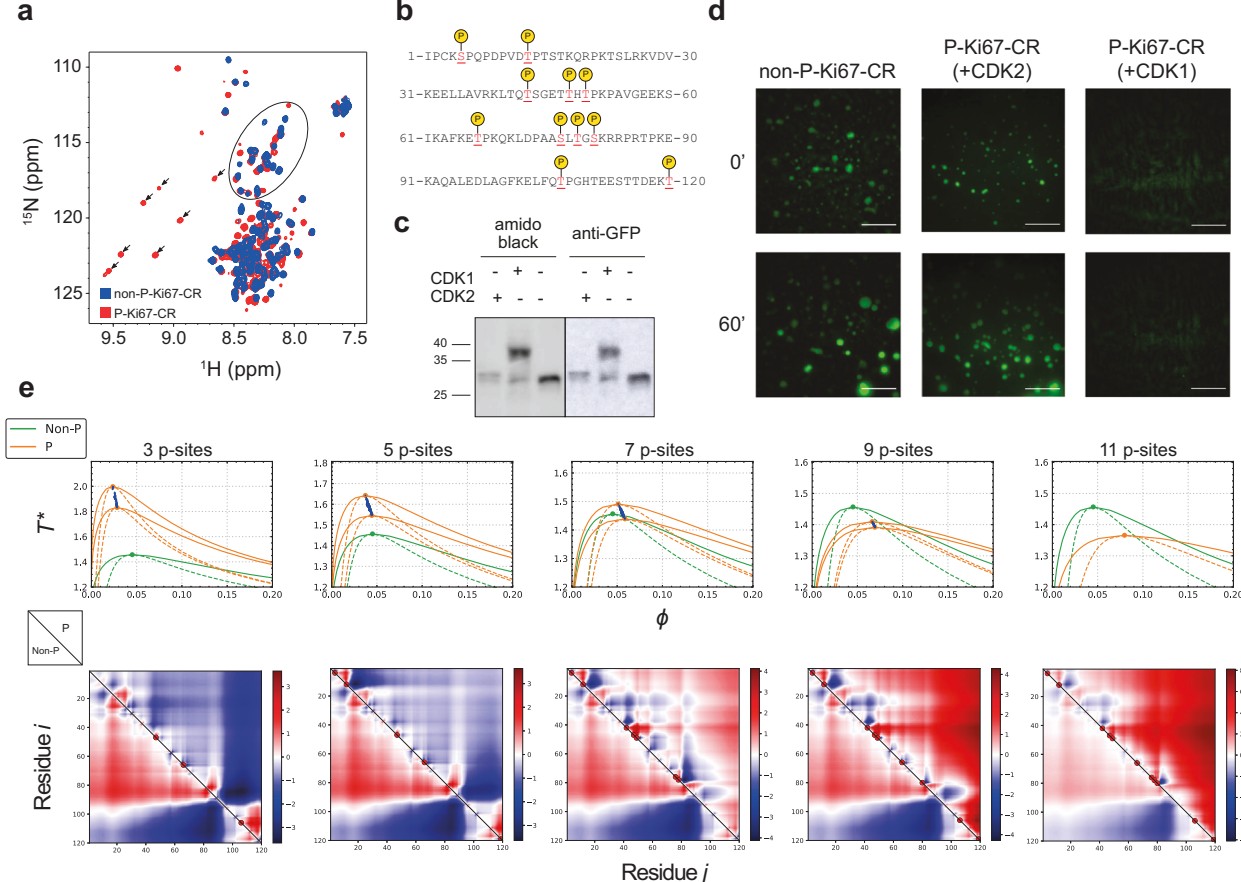

**Fig. 8 | CDK-mediated phosphorylation can generate a buffered phase separation switch. a** Overlaid NMR $^1$H-$^{15}$N HSQC of unphosphorylated (blue) and CDK-phosphorylated (red) GFP-tagged Ki-67 consensus repeat. Each cross-peak corresponds to one residue. The seven new deshielded cross peaks (highlighted by a black flag) appearing above 8.5 ppm in $^1$H correspond to phosphorylated serines or threonines ($^1$H downfield chemical shift perturbation on phosphorylated Ser/Thr residues due to phosphate electronegativity). Non-phosphorylated Ser/Thr residues are surrounded by a black oval. **b, c** GFP-Ki-67 consensus repeat was phosphorylated in vitro using recombinant CDK1-cyclin B-CKS1 or CDK2-cyclin A protein complexes and the phosphosites were mapped by mass-spectrometry (**b**) and the stoichiometry of phosphorylation was analysed by Phos-Tag SDS-PAGE (**c**) (amidoblack staining was used as loading control); $n = 1$ experiment. **d** Representative fluorescence images of in vitro phase separation assay with purified GFP-tagged Ki-67 consensus repeat (CR), non-phosphorylated (non-P) or in vitro phosphorylated (P) with recombinant CDK1-cyclin B-CKS1 or CDK2-cyclin A, at indicated time points; scale bars, 10 μm; $n = 2$ independent experiments, with three fields recorded for each condition. **e** Top, temperature-density phase diagrams for the consensus

repeat sequence of Ki-67. Critical temperature and density (blue circles) were computed for all possible 2048 sequences that arise from different degrees of phosphorylation. For a given degree of phosphorylation there are multiple possible sequences, of which two were chosen, corresponding to the highest and lowest values of the critical point. For these, temperature-dependent (in reduced unit) phase diagrams (solid orange) and spinodal lines (dashed orange) are shown along with the unmodified sequence (in green). Only critical points (blue circles) are presented for all the other sequences for a given stoichiometry/degree of phosphorylation. Bottom, SCDM maps of the unmodified sequence (Non-P, lower triangle) and a specific phosphorylated sequence (P, upper triangle). The phosphorylated sequence for a given stoichiometry (degree of phosphorylation) was chosen by selecting the sequence with the corresponding critical temperature and density closest to the average of the highest and lowest critical points. SCDM and Phase diagrams are consistent and show that phosphorylation can lower propensity to phase separate when eight or more sites are phosphorylated, contrary to sequences where six or less sites are phosphorylated. Source data are provided as a Source data file.

next performed in vitro phase separation assays and compared both partially and fully phosphorylated forms of the consensus repeat by varying incubation times and the activity of purified CDK complexes (Fig. 8c). The purified synthetic Ki-67 repeat could form droplets, and, as predicted, this was abolished upon full phosphorylation by CDK, whereas partial phosphorylation had no effect (Fig. 8d), suggesting that CDK-mediated phosphorylation acts as a buffered switch for phase separation. To define the properties of this switch, we analysed all 2048 combinations of the 11 phosphorylation sites on this synthetic protein by rG-RPA and SCDM. Strikingly, 1–6 phosphosites enhanced PS propensity while 8–11 phosphosites strongly reduced it (Fig. 8e, top), which was consistent with SCDM analysis (Fig. 8e, bottom). For low and high numbers of phosphosites, the overall behaviour was independent of the specific combination of phosphorylated sites, whereas 7 sites could either enhance or reduce PS propensity, depending on the exact combination. Taken together, our results lend support to our hypothesis whereby CDK-mediated phosphorylation can generate a buffered switch for homotypic interactions that contribute to PS.

## Discussion

In this multidisciplinary study, we show in vivo that CDK-dependent mitotic phosphorylation occurs synchronously on diverse proteins whose common denominators are a high level of disorder, localisation to MLOs, and multisite phosphorylation. Our data suggest that the majority of cell cycle-regulated phosphorylation may be controlling phase separation of components of membraneless organelles. Furthermore, CDK-mediated multisite phosphorylation may act as a phase separation switch promoting an abrupt transition into mitosis.

We first exploited the naturally synchronous cell cycles of *Xenopus laevis* to characterise cell cycle-regulated phosphorylation in single embryos at ultra-high resolution, allowing us to distinguish between progressive phosphorylation where different substrates become phosphorylated sequentially, as suggested by a model in which a complex combination of features encoded in the amino acid sequence of substrates determines phosphorylation timing[14], from the rapid phosphorylation that might result from the observed switch-like mitotic activation of CDK1[68]. A previous large-scale analysis of cell cycle-regulated phosphorylation used cell synchronisation procedures (nocodazole, thymidine) to show that potential phosphorylation sites are fully phosphorylated in mitosis[9], but phosphorylation dynamics could not be determined. True cell cycle synchronisation of entire cell populations has been deemed impossible due to differences between cells in the timing of entry into and exit from the blocks[22]. Thus, the rather progressive dynamics of CDK-mediated phosphorylation seen from synchronisation studies in fission yeast[69] may not reflect the behaviour occurring in single cells. Indeed, a comparison of cell cycle-regulated phosphorylation sites derived from synchronised cells[9] or from non-synchronised cells selected by flow cytometry[25] revealed that the synchronisation procedures used previously may artificially select for maximal phosphorylation. The same study[25] also showed that all identified phosphosites that peaked in mitosis displayed similar cell cycle behaviour, but the resolution was not sufficient to distinguish differences in kinetics between different phosphosites; this might be due to the elutriation windows used that, by definition, select similar but not identically sized cells from the population. Thus, our study in single embryos is unique to date, and demonstrates that all mitotic phosphosites observed undergo phosphorylation simultaneously, irrespective of the nature of the substrate. This is not incompatible with differences in affinity of CDK1 for different substrates, since a theoretical consideration of futile cycles demonstrates how graded inputs can generate all-or-nothing outputs[42]. We further show that this switch-like behaviour does not depend on regulated phosphorylation of CDK1 itself, consistent with the observation that tyrosine-15 phosphorylation of CDK1 is downregulated during early embryonic cell cycles[32,41]; instead, it can likely be accounted for by regulation by Greatwall of the phosphatase that reverses CDK-mediated phosphorylation, as well as by phosphorylation of NIPA, which inactivates the SCF ubiquitin ligase that degrades cyclin B in interphase[70].

Second, we reveal an underlying similarity between most cell cycle kinase substrates: they contain a much higher proportion of disordered residues than other phosphoproteins. While it has previously been found that intrinsic disorder generally predicts phosphorylation site localisation for any protein, irrespective of the kinase[43], and it was previously confirmed that CDK sites conform to this rule by tending to cluster in regions of proteins predicted to be disordered[6,44], what distinguishes cell cycle-regulated phosphorylation from other phosphorylation, and what determines which kinases are involved, has remained unknown. We find that substrates of CDKs, Aurora and Polo-like kinases are far more disordered than the phosphoproteome average. Even when substrates of two kinase families (CDK and MAP kinases) with the same minimal consensus site motif are compared, the CDK substrates have close to twice the disorder of MAP kinase substrates. This further highlights the enrichment for disordered proteins in cell cycle processes, noted recently[8]. Our data also resolve a circularity problem inherent in earlier work: previous observations of enrichment of CDK sites in disordered regions of proteins[6,44] might have a trivial explanation, since the amino acids constituting the consensus site for CDKs (S, T, P, K and R) are already highly enriched in disordered regions. Our bioinformatic approach corrects for this bias and confirms the validity of the previous observations. We propose that cell cycle kinases have been selected by evolution to phosphorylate the most highly disordered proteins of the proteome, and that this necessarily requires several families of kinases, to phosphorylate sites in disordered regions of proteins that are either positively charged at physiological pH (CDKs, Aurora) or negatively charged (PLK), while Aurora and PLK cannot phosphorylate proline-proximal sites.

Our observations that cell cycle kinase substrates are more disordered than other proteins and that they are frequently key components of MLOs lends the hypothesis that phosphorylation of these substrates might regulate MLOs themselves, consistent with the cell cycle-dependent assembly and disassembly of many MLOs. We provide a medium-throughput analysis of effects of phosphorylation on different CDK substrates that substantiates this hypothesis, which could only be achieved by applying a theoretical approach. Molecular dynamics simulations of biomolecules require large computer resources and timescales and are currently essentially unfeasible for modelling phase separation. In contrast, analytical simulations are powerful tools that can predict conformational properties of IDPs and average the ensemble of disordered states that depend on the sequence of IDRs, with high throughput and with minimal computer load[57]. Theoretical models also allow us to systematically vary different inputs, such as post-translational modifications, mutations, salt concentrations and pH to explore their relative contributions. Our calculations with one IDP and solvent predict binary phase diagrams, and the results match our experimental data for the protein that we use to test our hypothesis, Ki-67.

A previous study provided evidence for phospho-regulation of MLOs in mitosis but attributed it to a different kinase, DYRK3[45]. However, DYRK3 inhibition led to mitotic formation of abnormal hybrid condensates that contained material from various MLOs (splicing speckles, stress granules and pericentriolar material), but did not prevent the normal breakdown of MLOs such as nucleoli, nuclear pore complexes or Cajal bodies. Our data rather implicate CDK1 as the likely major MLO regulator in mitosis. This is not unprecedented since CDKs can prevent phase separation of components of replication complexes[71], promote nuclear pore disassembly[72] and dissolve stress granules in yeast[73]. We additionally provide evidence that CDK-mediated phosphorylation can both promote or inhibit biological

phase separation, depending both on the sequence context and the stoichiometry of phosphorylation. The latter adds an additional switch-like regulation to the onset of mitosis: passing a threshold number of phosphorylations can switch from promoting phase separation to inhibiting it. Thus, our data show that CDK-mediated multisite phosphorylation may act as a buffered switch for phase separation, which is independent of the exact combinations of phosphorylated sites, providing a robust underlying mechanism that may contribute to the abruptness of the cellular reorganisation at the entry into mitosis. Interestingly, phosphosites specific to meiotic M-phase samples appeared to be the most highly phosphorylated, and the most multiphosphorylated peptides were detected in these samples. We speculate that such a prevalence of "hyper-phosphorylation" might reflect the long-term maintenance of the M-phase state in unfertilised eggs, and suggest that a more detailed quantitative analysis of the threshold phosphorylation levels required to generate and maintain meiotic versus mitotic M-phase might reveal interesting differences between these two states.

Finally, we provide evidence that the perichromosomal layer of mitotic chromosomes may be liquid, and we suggest a mechanism for mitotic targeting of nucleolar components to the perichromosomal layer by Ki-67[59,60] via CDK-mediated phosphorylation, which reduces PS propensity of several major nucleolar IDPs, thus triggering nucleolar disassembly, while simultaneously promoting PS of Ki-67 to recruit nucleolar components to chromosomes.

## Methods
### Egg collection and in vitro fertilisation
All animal procedures and experiments were performed in accordance with national animal welfare laws and were reviewed by the Animal Ethics Committee of the Royal Netherlands Academy of Arts and Sciences (KNAW). All animal experiments were conducted under a project license granted by the Central Committee Animal Experimentation (CCD) of the Dutch government and approved by the Hubrecht Institute Animal Welfare Body (IvD), with project licence number AVD80100201711044. Female *X. laevis* frogs (aged >2 years; Nasco, catalog N° LM00535) were primed with 50 international units (IU) of human chorionic gonadotropin at least 2 days, and no more than 7–8 days before a secondary injection with 625 IU to induce ovulation. Roughly 16 h after the second injection, fresh eggs were collected by pelvic massage and kept in 1x Marc's Modified Ringer's (MMR). Next, eggs were placed in a petri dish and checked under the microscope to keep only those that exhibited the healthy pigment pattern (dark animal pole and white vegetal pole).

To perform the in vitro fertilisation, around 1/3 of a full testis (prepared from adult *X. laevis* male frogs, purchased from the European Xenopus Resource Centre) was cut into fine pieces and mixed with in 500 μl of 1x MMR. The suspension was pipetted up and down until big clumps were dissolved. Next, buffer was removed from the petri dish and the eggs were collected. Once eggs were well dispersed across the dish, the sperm suspension was added. The dish was then flooded with 0.1x MMR to induce fertilisation.

### Sample collection
The first time point was collected immediately before adding the sperm suspension (time 0' corresponds to the unfertilised egg). Eggs were kept at room temperature (18–20°) and under a dissecting microscope after fertilisation. At approximately 15 min, fertilised eggs underwent shrinkage of the animal hemisphere and rotation within the vitelline membrane, so that the animal hemisphere faced upwards. These changes are known indicators of successful fertilisation, so only the eggs that underwent these changes were used for the experiment.

Samples were collected approximately every 15 min. Eggs were rapidly placed in individual tubes and snap-frozen in liquid nitrogen, trying to preserve the phosphorylation events occurring at that specific time. Since the eggs were monitored under the microscope, we were able to determine if samples were collected before or after a cell division had occurred.

### Xenopus egg extracts
For frog extract preparation, adult *Xenopus* females were obtained from the Centre de Ressources Biologiqes Xénopes (CRB; Rennes, France), and housed in a dedicated aquatic facility located at the CRBM (Approval N° B34-172-39). Animals were used following regulations according to the Direction Générale de la Recherche et Innovation and the French Ministry of Higher Education, Research and Innovation. All procedures were validated by the animal welfare committee of the Occitanie region.

Interphase *Xenopus* egg extracts were prepared, and DNA replication time courses performed, as described previously[74]. In brief, eggs laid overnight in 150 mM NaCl were dejellied in degellying buffer (29 mM Tris pH 8.5, 110 mM NaCl, 5 mM DTT); rinsed several times in High Salt Barths solution (15 mM Tris pH 7.6, 110 mM NaCl, 2 mM KCl, 1 mM MgSO$_4$, 0.5 mM Na$_2$HPO$_4$, 2 mM NaHCO$_3$), twice in MMR (5 mM HEPES-KOH pH 7.6, 100 mM NaCl, 2 mM KCl, 0.1 mM EDTA, 1 mM MgCl$_2$, 2 mM CaCl$_2$), before activation with 0.3 μg/ml calcimycin ionophore in MMR. Subsequently, two rinses in MMR and two more in SB (50 mM HEPES-KOH pH 7.6, 50 mM KCl, 2.5 mM MgCl$_2$, 5% Sucrose, 0.014% β-mercaptoethanol) followed, while during the last rinse the eggs were transferred on ice and SB was supplemented with protease inhibitors (10 μg/ml leupeptin, pepstatin and aprotinin). Eggs were spun down at 200 g for 1 min and excess of buffer removed before being centrifuged at 16,000 × $g$, 4 °C for 10 min. Protease inhibitors and 10 μg/ml cytochalasin B were added to the cytoplasmic fraction. Extracts were further centrifuged in SW55Ti rotor for 20 min at 20k rpm (48,000 × $g$) at 4 °C. The cytoplasmic layer was extracted with a large-bore needle and syringe, and supplemented with glycerol 3% and ATP regenerating system (10 mM creatine phosphate, 10 μg/ml creatine kinase, 1 mM ATP, 1 mM MgCl$_2$) added from a 20x stock. Aliquots were frozen in liquid nitrogen. Mitosis was induced in extracts by adding recombinant GST-Cyclin BΔ90 (40 ng/ml).

### Cell lysis
For the cell lysis, we used a similar approach to that described by Lindeboom et al.[75]. Briefly, each sample was thawed and homogenised with 15 μL of ice-cold cell lysis buffer (20 mM Tris-HCl pH 8.0, 70 mM KCl, 1 mM EDTA, 10% glycerol, 5 mM DTT, 0.125% Nonidet P-40, 1 mM PMSF, 1x complete EDTA-free protease inhibitor, 1xPhoStop). Samples were subsequently centrifuged at max speed on a benchtop Eppendorf centrifuge. 10 μL of soluble material was recovered and snap-frozen in liquid nitrogen. Samples were stored at −80 °C until further processing.

### Protein digestion and phosphopeptide enrichment
Cell lysates were digested using the FASP method[76]. Briefly: proteins were thawed and immediately reduced and alkylated with 10 mM DTT and 0.05 M iodoacetamide. Next, proteins were digested with Lys-C (overnight) at 37 °C in a wet chamber, followed by addition of trypsin and further incubation under the same conditions for 4 h. Both enzymes were used at 1:50 enzyme to protein ratio (protein quantification by Bradford assay showed that each individual egg provides ~20 μg of yolk-free protein). For the egg extract experiment, each FASP filter was loaded with 200 μg of protein. Peptides were cleaned using the Oasis HLB 96 well plates (Waters Corporation) and consequently subjected to phosphopeptide enrichment using Fe(III)-NTA 5 μL cartridges in the automated AssayMAP Bravo Platform (Agilent Technologies), as described by Post et al.[28]. Both flow through (peptides) and eluates (phosphopeptides) were dried down and stored at −80 °C until further use.

## LC-MS/MS analysis

All samples for label-free shotgun proteomics were analysed using a UHPLC 1290 system (Agilent Technologies) coupled to an Orbitrap Q Exactive HF mass spectrometer (Thermo Fisher Scientific). Nano flow rate was achieved using a split flow setup aided by an external valve as described by Meiring et al.[77]. Peptides were first trapped onto a pre-column (inner diameter [ID] of 100 μm and 2 cm length; packed in-house with 3 μm C18 ReproSil particles [Dr. Maisch GmbH]) and eluted for separation into an analytical column (ID of 75 μm and 50 cm length; packed in-house with 2.7 μm Poroshell EC-C18 particles (Agilent Technologies). The latter was done using a two-buffer system, consisting of buffer A (0.1% formic acid [FA] in water) and buffer B (0.1% FA in 80% ACN). Peptides were trapped during 5 min at 5 μL/min flow-rate with solvent A. For the measurement of the full proteome, we used a 155 min gradient from 10 to 36% of solvent B. For the phosphoproteome, we used a 95 min gradient from 8 to 32% of solvent B. Both methods included a wash with 100% solvent B for 5 min followed by a column equilibration with 100% solvent A for the last 10 min.

The mass spectrometer was operated in data-dependent acquisition (DDA) mode. Full scan MS was acquired from 375–1600 $m/z$ with a 60,000 resolution at 200 $m/z$. Accumulation target value was set to 3e6 ions with a maximum injection time of 20 ms. Up to 15 (12 for the phosphoproteome) of the most intense precursor ions were isolated (1.4 $m/z$ window) for fragmentation using high energy collision induced dissociation (HCD) with a normalised collision energy of 27. For MS2 scans an accumulation target value of 1e5 ions and a maximum injection time of 50 ms were selected. Scans were acquired from 200–2000 $m/z$ with a 30,000 resolution at 200 $m/z$. Dynamic exclusion was set at 24 s for the proteome and 12 s for the phosphoproteome.

For targeted proteomics, an EASY-nLC 1200 System (Thermo Fisher Scientific) coupled to an Orbitrap Q Exactive HF was used. Peptides were separated using an EASY-Spray analytical column (ID of 75 μm and 25 cm length; packed with 2 μm C18 particles with a 100 Å pore size) (Thermo Fisher Scientific). Gradient lengths were shortened to 60 min. Phosphopeptides of interest from the previous experiment were selected and heavy-labelled versions were synthesised (JPT Peptide Technologies). These synthetic standards were used during method development for retention time scheduling and instrument ion fill-time optimisation. Additionally, synthetic heavy peptides were pooled and combined with synthetic retention time peptide standards (iRT, Biognosys) to generate a spectral library, measured in DDA mode using the same LC-MS setup. This spectral library provided fragment intensity and retention time information for quality control assessment of targeted measurements. Samples were reconstituted in 2% FA containing ~200fmol of each synthetic standard. The mass spectrometer was operated in data-independent acquisition mode with an inclusion list of targets for parallel reaction monitoring (PRM). The list included the $m/z$ values for the heavy and light versions of the phosphopeptides. Optimal measurement parameters were determined using test samples spiked with the heavy-labelled standards in order to guarantee optimal sensitivity for detection of endogenous phosphopeptides. We measured the targets of interest in a scheduled fashion, during a four-minute window with a 120,000 resolution, maximum injection time of 246 ms and an accumulation target value of 2e5 ions, to ensure maximum specificity and sensitivity.

## Data processing

DDA raw files were processed with MaxQuant[78] (v1.6.0.1) using a false discovery rate (FDR) < 0.01. The default settings were used, with the following exceptions: variable modifications, specifically methionine oxidation, protein N-term acetylation and serine, threonine and tyrosine phosphorylation were selected. Cysteine carbamidomethylation

was selected as a fixed modification. We also enabled the 'match between runs' option with the default values. Fractions were set so that matching was done only among biological replicates and samples of consecutive time points. The database search was conducted against a database generated by Temu et al.[79]. This was particularly helpful since other publicly available databases contained several incomplete and/or poorly annotated sequences, which proved to be impractical for further data analysis.

The data was uploaded to the Perseus platform[80] for further analysis. Briefly: decoy sequences and potential contaminants were filtered out. Only high confidence localisation (>0.75 localisation probability) phosphosites were conserved for further analysis. Intensities were $\log_2$ transformed and then normalised by subtracting the median intensity of each sample. Biological replicates were grouped accordingly by time point; this grouping allowed us to filter the data and keep only those phosphosites that could be detected in at least two out of three biological replicates in any of the time points. Missing values were imputed from a random normal distribution applying a downshift of 1.8 times the standard deviation of the dataset, and a width of 0.3 times the standard deviation. This effectively replaced missing values at the lower end of the intensity distribution. We then performed an ANOVA (Benjamini–Hochberg FDR < 0.05) to determine which phosphosites displayed statistically significant changes through the time course. Average phosphosite intensities were grouped using a combination of k-means and hierarchical clustering using the ComplexHeatmap package[81] in R. Protein intensities were processed in a similar fashion, removing proteins that were only identified by peptides that carry one or more modified amino acids.

Next, the full list of proteins with significantly changing phosphosites were matched against the human Uniprot database using the Basic Local Alignment Search Tool (BLAST), to render Uniprot identifiers that were compatible with different Gene Ontology (GO) analysis tools. We used the STRING web tool[82] to gain insight into the relation amongst dynamically phosphorylated proteins. The full list of phosphoproteins was uploaded and analysed using default settings. Next, the protein network was loaded into Cytoscape for clustering and visualisation. Proteins were clustered using GLay community clustering[83] and enrichment of GO terms per cluster was obtained using BiNGO[84] (shown in Fig. 1b).

GO term enrichment was also acquired individually for each group (A–D) obtained after hierarchical clustering of dynamic phosphosites. For this we used STRING and filtered the enriched GO terms to keep only those with $p < 0.01$, fold enrichment >5 and a minimum of 5 proteins per term. The list of terms was further condensed by removal of redundant terms using the Revigo web tool[85]. Remaining GO terms (including BP, MF and CC) were manually curated to further avoid redundancy. The final set of GO terms per cluster are shown in Supplementary Fig. 1. Following the same strategy, we analysed GO term enrichment for the interphase and mitotic clusters from the in vitro dataset separately (shown in Supplementary Fig. 3).

PRM raw files were analysed with Skyline software[86]. Signal quality for each target of interest was assessed visually for all samples. Quality control of endogenous signals was done by confirming the perfect co-elution of both peptide forms (heavy and light), assessing their retention time and peak shape. We also used the similarity of the relative intensity of fragment ions (rdotp > 0.9) between light and heavy to exclude signals that showed poor correlation. Quantifications were done with a minimum of three fragments per phosphopeptide. The data was loaded into R for data cleanup and visualisation, using the Complex-Heatmap and ggplot2[87] packages.

## Motif analysis

Obtained phosphopeptides were aligned by centering them around the phosphosite detected and the conserved motifs for the different

kinases were determined using regular expressions by applying the following rules:

PLK: [D/N/E/Y]-X-[S/T]-[Hydrophobic / ^P]
AURA/AURB: [K/R]-X-[S/T]*[^P]
NEK: [L|M|F|W]-X-[S|T]*-[A|V|I||L|F|W|Y|M]-[K|R]
Casein kinase 1: [D/E]-[D/E]-[D/E]-X-X-[S/T]* or [S/T]-X-X-[S/T]*
Casein kinase 2: [S/T]-[S/T]*-X-[E/D/S]
DDK: [S/T]*-[E/D]-X-[E/D] or [S/T]*-[S/T]-P
PKA: R-[R/K]-X-[S/T]*-[Hydrophobic]
Cdk full consensus motif: [S/T]*-P-X-[K/R]
Cdk minimal consensus motif: [S/T]*-P

Where [ ] groups multiple amino acids for one position, ^ at the left of a certain amino acid informs that it is forbidden for that position, and X represents any amino acid.

## Data collection for human and yeast CDK1 targets

Data of CDK1 substrates for *S. cerevisiae* were downloaded from online supplementary information of papers describing two different studies using in vitro (4) and in vivo (26) approaches, respectively. We defined high confidence yeast CDK1 targets as the intersection of both datasets. Other phosphorylations detected in both studies for which there was no evidence for CDK1 involvement were considered as the non-CDK1-mediated phosphoproteome (universe). For human CDK1 subfamily targets, we extracted information available in the PhosphoSitePlus database[88]. An additional step of manual curation from the following studies[4,5,62,89–104] was performed to obtain a high confidence human CDK1 subfamily targets dataset. The phosphoproteome universe was constructed with all the phosphorylated proteins deposited in the PhosphoSitePlus database after subtraction of the CDK1 subfamily targets. Data is available in Supplementary Data 2.

## Data collection for MLO proteomes

Data from proteomics studies of the composition of MLOs characterised by liquid-liquid phase separation was obtained from the following sources: stress granules[105,106], nuclear speckles[106,107], PML nuclear bodies[106,108], P-bodies[109], nucleoli[110,111], nuclear pore complexes[112], Cajal bodies[106,113], Super-enhancer-Mediator condensates[114]. Data is available in Supplementary Data 4.

## Prediction of intrinsically disordered regions

For the UniProt proteomes of human, yeast and *Xenopus laevis*, disorder information was fetched from MobiDB[115] with the exception of SPOT disorder predictor, which was calculated for all the proteins of each dataset. For *Xenopus* proteomics studies, we used the available standalone software of IUPred, VSL2B, and SPOT to predict IDRs in all the proteins of the database. Data is available in Supplementary Data 3.

## Differential disorder composition

For the three organisms analysed (*Xenopus*, human and yeast), the amino acid composition for the entire phosphoproteome and for the disordered regions of the phosphoproteome was calculated. For each amino acid, we estimated the differential disorder composition with the equation:

$$(Comp. Disorder - Comp. Phosphoproteome)/Comp. Phosphoproteome. \tag{1}$$

Positives values show amino acids enriched in disordered regions while negative values represent amino acids depleted in disordered regions.

## Bioinformatic and statistical analysis of disorder and phosphorylation

All the statistical analysis was performed with the R programming language (https://www.r-project.org/) using R studio as an integrated development environment (available at https://rstudio.com/). The packages Tidyverse[87] and Bioconductor[116] were used for cleaning, manipulation, and graphical representation of the data. Sequence logos were generated using the information content as described[117]. IUPred scores were plotted with an ad hoc designed script, available upon request.

For the contingency table analysis, the disordered regions of CDK targets and dynamic phosphoproteins were calculated with three predictors (IUPred, VSL2B, and SPOT). For each combination of disorder predictor and phosphorylation dataset, a two-by-two table with the counts of phosphorylatable residues (Ser/Thr) phosphorylated or not, either located in IDRs or in structured regions (Supplementary Fig. 6B), was generated. Each table was then analysed with the Fisher test for obtaining the odds ratio and the associated *P*-value.

Codes for phosphorylation and disorder analysis developed in this study are uploaded in Github [https://github.com/gero007/CDK-mediated-phosphorylation-of-IDRs].

## Calculation of SCDMs

Elements (i,j) of SCDM were calculated using Eq. 1 of Huihui and Ghosh[56]. In this coarse grain model, each amino acid is considered a point with a charge q = −1 for Aspartic and Glutamic acid, charge q = 1 for Arginine and Lysine, charge q = 0 for all other amino acids. Phosphorylation is modelled by replacing neutral charge of Serine, Threonine to q = −2 to mimic the effect of double negative charge of phosphate groups. SCDM maps were made visually continuous between neighbouring (i,j) pairs using spline-16 interpolation.

## Calculation of rG-RPA

Theoretical predictions of phase diagrams were made using a newly developed theory called renormalised-RPA (rG-RPA)[54]. This theory explicitly accounts for sequence of charges in the protein and builds temperature-dependent phase diagrams yielding coexistence curves of protein-dense and protein-dilute phases. At the present time, rG-RPA is the only mathematical theory that reproduces known experimental and simulation results for both polyampholytes and polyelectrolytes as a function of sequence charge (not just the composition). It is an approach based on Hamiltonian (energy function) and hence do not invoke any ad-hoc charge patterning terms and directly predicts full phase diagram. We used the sequence of charges following the same protocol used for calculating SCDM for the phosphorylated and non-phosphorylated proteins. In addition to the entropic and electrostatic interaction terms used in rG-RPA, a generic Flory-Huggins type of mean-field interaction:

$$\chi = \varepsilon/kT \tag{2}$$

term was added to the free energy. This interaction term models hydrophobic, cation-pi and all other non-electrostatic interactions that may also drive phase separation. A common value of $\varepsilon = 2\,kT$ was used for all sequences for simplicity. The phase diagrams are given in terms of a dimensionless temperature $T^*$. It is not possible to predict exact temperatures of phase transition. However, the relative propensity to phase separate between two sequences can be predicted by comparing phase diagram in this non-dimensional temperature unit. Since the goal of the study is to investigate the electrostatic role of phosphorylation (induced by CDK) in phase separation, we used the same value of e for both the phosphorylated and non-phosphorylated sequences. The generic choice of $\varepsilon = 2\,kT$ is reasonable based on transfer energy of solvophobic amino acids[118], and any other non-electrostatic interaction such as pi-pi, cation-pi that may drive phase separation. For Ki-67 consensus sequence and individual repeat units, we used a slightly higher value of $\varepsilon = 2.5\,kT$ since experiments required crowding to induce phase separation. Overall trends are insensitive−even though the relative degree of differences in phase separation propensity may

change—if higher values of ε are used, indicating robustness of the predictions of increased or suppressed propensity to phase separate due to CDK-mediated phosphorylation. Codes for rG-RPA calculation are uploaded in Github [https://github.com/MaxCalLab/IDPTheory/tree/main/rG-RPA].

## Selection of IDRs for SCDM and rG-RPA calculations

Intrinsically disordered region (IDR) selection for each protein was accomplished using the following protocol. Regions of high disorder were identified primarily using the AlphaFold 2 (AF2) predicted local-distance difference test (pLDDT)—a measure of confidence on the residue level[58,119,120]. Recent work has suggested that residues with very low pLDDT scores (pLDDT <50) are likely to be disordered[121]. In cases where no published structure existed in AF2, the IUPred2A score was used and IDRs were chosen as the longest region with a score above 0.5 (scores above 0.5 indicate intrinsic disorder)[122,123]. For proteins with an AF2 entry, the choice of IDR was made for the longest region where the majority of residues (more than 60%) had pLDDT scores below 50. This was done by extracting raw pLDDT values from the mmCIF files available for each protein entry on AF2. Further refinement of IDR selection can be accomplished by finding the longest ordered region (LOR), i.e. the longest stretch of consecutive residues with pLDDT values above 50, within the IDR. If the individual LOR represented more than 15% of total residues within the IDR, the LOR and following IDRs were not included in the parent IDR assignment. The choice of 15% used as a threshold is arbitrary and may be adjusted.

## Coarse-grained force field

We adopted a one-bead-per-residue coarse-grained (CG) model that has been shown to capture the structural and phase separation properties of flexible proteins as a function of their sequence[124] and phosphorylation pattern[125]. In this framework, bonded interactions between neighbouring residues were modelled using a harmonic potential with a constant of 1000 kJ/mol/nm² and a bond length of 0.38 nm. Electrostatic interactions between charged residues, i.e. Asp, Glu (−1e); Lys, Arg (+1e); His (+0.5e); ph-Ser, ph-Thr (−2e), were computed with a screened Coulomb potential, using a Debye-Huckel length of 1 nm. Short-range interactions were modelled by Lennard-Jones (LJ) potentials defined by

$$V_{ij}(r) = 4\lambda_{ij}\epsilon\left((\sigma_{ij}/r)^{12} - (\sigma_{ij}/r)^6\right) \qquad (3)$$

where values of $\lambda_{ij}$, $\epsilon$, and $\sigma_{ij}$ used reported values[124,125] with the exception of $\lambda_{ArgArg}$ that was set equal to 0.01, instead of 0.00, in order not to neglect excluded volume effects while using LJ functional form. In the simulation of full-length Ki-67, the conformation of the N-terminal folded domain (1–128) was restrained by an elastic network based on experimental structure (PDB: 1R21) with a distance cutoff of 2 nm and an elastic constant of 5000 kJ/mol/nm². All MD simulations were performed with GROMACS 2018.3[126]. Simulations were run in the NVT ensemble, controlling the temperature by means of a Langevin thermostat with a friction constant of 25 ps⁻¹ and a time step of 10 fs.

## Single-chain MD simulations of full-length Ki-67

The structure with PDB id 1R21 (https://www.rcsb.org/structure/1r21) was used to model the conformation of the first 128 residues of the full-length Ki-67, while TraDES[127,128] was used to generate the initial conformation of the remaining disordered part of the non-phosphorylated molecule. The models were then joined using UCSF Chimera 1.14[129]. The initial configuration for the phosphorylated full-length Ki-67 was obtained by 'mutating' (i.e. using parameters that have been tuned specifically for ph-Ser/ph-Thr) the amino acids in the non-phosphorylated structure. Initial configurations were inserted in a cubic simulation box with a side of 100 nm with periodic boundary

conditions. We employed a Parallel Tempering (PT-MD) scheme, with eleven replicas in the 300–400 K range in order to enhance the conformational sampling and obtain a reliable estimate of the radius of gyration of the protein as a function of the temperature. PT-MD simulation ran for 1e8 steps, attempting Monte Carlo exchanges between neighbouring replicas every 1e2 steps and saving snapshots every 1e4 steps for further analysis. This simulation length is sufficient to converge the observables of interest, in particular the radius of gyration of the polymer chains (see Supplementary Fig. 10). Data are provided in Supplementary Data 6.

## Single-chain MD simulations of Ki-67 consensus repeat

The initial extended configuration of the non-phosphorylated consensus repeat of Ki-67 was generated with the tLEAP tool available in AmberTools18[130] while the phosphorylated structure was generated by 'mutating' (i.e. using parameters that have been tuned specifically for ph-Ser/ph-Thr) the residues in the non-phosphorylated configuration. PT-MD simulation followed the same protocol employed for the simulations of full-length Ki-67, with the only difference being the range of temperatures, between 200 K and 500 K, with intervals of 10 K. Data are provided in Supplementary Data 6.

## Phase coexistence MD simulations

Phase coexistence simulations of monomers and dimers of the non-phosphorylated consensus repeat and of monomers of the phosphorylated consensus repeat were carried out employing the slab method[124]. Initial configurations of the slab simulations for each system were generated by inserting 200 copies of the respective molecules in a 20 × 20 × 30 nm box in random positions and orientations with the *gmx insert-molecules* tool available in GROMACS 2018.3[126], and extending the simulation box in the z direction to 200 nm. Simulations were run for 1.5e8 steps, saving frames every 10,000 steps for further analysis, discarding the first 0.5e8 steps of the simulations as equilibration. This simulation length is sufficient to equilibrate the density profiles along the z coordinate of the system used to evaluate the equilibrium densities (see Supplementary Fig. 11). Data are provided in Supplementary Data 6.

## Estimation of theta temperatures

Coil-to-globule transition temperatures ($T_\theta$) were estimated for non-phosphorylated full-length Ki-67 and the monomer of the non-phosphorylated consensus repeat from single chain PT-MD simulations, following a reported approach[131]. Intramolecular distances as a function of the chain separation $|i–j|$ were computed for each temperature replica and fitted with the following expression:

$$R_{ij} = 0.55|i-j|^\nu \qquad (4)$$

where the scaling exponent $\nu$ is the fitting parameter. We then determined $T_\theta$ as the temperature corresponding to $\nu = 0.5$.

## Estimation of binodal curves

Density profiles from phase coexistence simulations were estimated by means of the gmx density tool available in GROMACS 2018.3, centering the dense phase in the middle of the z-axis. The concentration of the diluted phase was computed by averaging the density in the box at 0 nm < z < 60 nm and 140 nm < z < 200 nm, while the concentration of the dense phase was obtained by averaging the density at 90 nm < z < 110 nm. Following the approach of[124] the critical temperatures, $T_c$, for the three systems investigated with phase coexistence simulations, have been evaluated by fitting the following equation:

$$\rho_H - \rho_L = A(T_c - T)^{0.325} \qquad (5)$$

## Human opto-Ki-67 plasmid construction

pCDNA5_FRT_Ki67-FL(or repeats+LR or dLR)-mCherry-Cry2 was generated by PCR amplification of human Ki67 from pcDNA5-FLAG-GFP-Ki-67 plasmid[59]. The following primers were used for the cloning:

full-length Ki-67:

1st fragment (2501 bp) F1 Fwd taagcttggtaccatcgatgatgtggcccacgagacgcc F1 Rev ttctctaatacactgccgtcttaagggagg

2nd fragment (5314 bp) F2 Fwd gacggcagtgtattagagaaaatggaaacgtagc F2 Rev tgggacgtgtcttggggcatctctttgtg

3rd fragment (1955bp) F3 Fwd atgccccaagacacgtcccaggaaagaag F3 Rev tatcctcctcgcctttagacaccatgataatatcttcactgtccctatgac

Repeats+LR construct:

1st fragment (2501 bp) F1 Fwd taagcttggtaccatcgatgatgtggcccacgagacgcc F1 Rev ttctctaatacactgccgtcttaagggagg

2nd fragment (6285 bp) F2 Fwd gacggcagtgtattagagaaaatggaaacgtagc F2-r+LR Rev tatcctcctcgcctttagacaccatgattggtgtttgagagagctcttggaagctg

dLR construct:

1st fragment (4812 bp) dLR-F1 Fwd taagcttggtaccatcgatgatgactaaaatgccctgccagtcattacaacc F2 Rev tgggacgtgtcttggggcatctctttgtg

2nd fragment (1955bp) F3 Fwd atgccccaagacacgtcccaggaaagaag F3 Rev tatcctcctcgcctttagacaccatgataatatcttcactgtccctatgac

The three fragments were cloned into the AflII/KpnI digested pCDNA5_FRT_TO_TurboID-mCherry-Cry2 (Addgene plasmid # 166504) using the In-Fusion HD Cloning Kit protocol.

## Generation of Flp-In T-REx 293 opto-Ki67 cell lines

Flp-In T-REx 293 (Termo Fisher Scientific, Darmstadt, Germany) cell line was grown under standard conditions (37 °C, 5% $CO_2$) in Dulbecco's modified Eagle's medium (Merck-Sigma-Aldrich, D5796). The medium was supplemented with 10% foetal bovine serum (FBS), 100 µg/ml Zeocin and 15 µg/ml Blasticidin. One million HEK-293 T-REx cells were plated in a 6-well plate 24 h before transfection. The next day, 500 ng of each optoKi67 expression plasmid is combined with 3.5 µg pOG44 encoding the Flp recombinase. Transfection was made with 8 µl Lipofectamine 2000 according to the manufacturer's instructions. 48 h post transfection, cells were transferred to a 100 mm petri dish. On the next day, selection was performed by adding hygromycin B at a final concentration of 50 µg/mL. Around 14 days after selection, clones were pooled and expanded. The cells were tested for the expression of the construct by immunofluorescence. Flp-InT-Rex 293 opto-Ki67 stable cell lines were maintained with 15 µg/mL Blasticidin and 15 µg/mL Hygromycin.

## Overexpression of GFP-Ki-67

HeLa cells were plated at the density of $150 \times 10^3$ cells per well in 6-well plate, containing 10 mm coverslips. Cells were transfected the following day with GFP-Ki-67[59] using FuGene reagent following the manufacturer's instructions, at 3:1 FuGene/DNA ratio. Cells were fixed with 3.7% formaldehyde 24 h or 48 h later, stained with the indicated antibodies and mounted on slides with Prolong Gold with DAPI (Invitrogen, #P36935). Images were taken with Leica Thunder upright microscope using a 100x objective, with concurrent Computational Clearing.

## Fluorescence recovery after photobleaching (FRAP)

HeLa cells were seeded in glass bottom plates and 24 h after they were transfected with a GFP-tagged Ki-67 plasmid[59] with lipofectamine 2000 in Optimem serum-reduced medium following the manufacturer's recommendations. FRAP was performed 24 h or 48 h after transfection in a confocal Zeiss LSM980 NLO microscope at 37 °C and 5% $CO_2$. Image acquisition was made with 488 nm laser and photobleaching was performed with the same laser at 75–80% power. Multiple regions per cell were photobleached. Two images were acquired

pre-photobleaching and every 1 s for 100 s. Average fluorescence intensity values over time were obtained for each region, for the corresponding contiguous unbleached regions and for the nucleus, with the FIJI plugin "create spectrum jru v1". Cells and/or regions that moved were excluded from the analysis. Analysis was continued in R. Correction for non-intentional acquisition photobleaching was performed by dividing intensity values of the photobleached region or the contiguous unbleached region by the average of the whole nucleus for each time point. Normalisation was performed by setting the intensity values pre-photobleaching (time = −1s) as the maximum. Normalised average fluorescence recovery curves of the photobleached regions and unbleached regions of each cell were plotted in the same graph. The average intensity values over time of the photobleached regions were fitted with the following non-linear regression equation from the frapplot library in R:

$$y = c - a \cdot e^{b \cdot t} \qquad (6)$$

Where:

$c$ = ymax = maximal intensity value
$c-a$ = ymin = minimal intensity value

$$b = -k$$

$$T - half = \frac{1}{k \cdot \ln(2)} \qquad (7)$$

$$\text{Total recovery} = TR = \frac{ymax - ymin}{1 - ymin} \qquad (8)$$

Each cell was classified by the type of Ki-67 expression: mitosis (perichromosomal layer), interphase (low-medium levels of Ki-67 expression) and high expression in interphase. The half-time and percentage of recovery values were plotted in R with the ggplot2 library. Statistical significance was calculated with the Wilcoxon test in R.

## Opto-Ki-67 activation

Cells were plated in DMEM on coverslips a day prior to activation. Expression of opto-Ki67 was induced with 2 µg/ml doxycycline for 16 h. For light activation, plates were transferred into a custom-made illumination box containing an array of 24 LEDs (488 nm) delivering 10 mW/cm² (light intensity measured using a ThorLabs-PM16-121-power meter). Cry2 activation was induced using 4 min of blue light cycles: 4 s On followed by 10 s Off. Cells were fixed with 4% paraformaldehyde (PFA) for 15 min at RT, counterstained with Hoechst (Invitrogen, Cat H21491) in PBS-TritonX (0.2%), and mounted on glass slides using Prolong Gold antifade reagent (Invitrogen, Cat P36930). Where indicated, cells were counterstained with antibodies and images were captured using a 63x objective (NA 1.46 oil).

To check the effect of inhibitors on Ki-67 foci formation, the cells were incubated for 1 h with 0.5 µM okadaic acid, 5 µM purvalanol A or vehicle (DMSO) for 1 h prior to light activation and fixation. Foci number was analysed using FIJI Software and statistical significance was assessed by one-way ANOVA on ranks (Kruskal−Wallis test) and pairwise post hoc comparisons using the Mann−Whitney test. P-values were adjusted by the Benjamini−Hochberg method. Plots were generated using the ggplot2 library in R.

## Ki-67 consensus repeat DNA construct

The cDNA sequence coding for Ki-67 consensus repeat (Ki-67-CR) was ordered from IDT® gene synthesis. It was subsequently cloned into pDB-GFP plasmid between HindIII and XhoI sites to obtain the pDB-GFP-Ki-67-CR vector. In this construct, Ki-67-CR was fused with a (his)

6-GFP N-terminal tag. GFP sequence is followed by the HRV 3C (3C) protease recognition site (Leu-Glu-Val-Leu-Phe-Gln/Gly-Pro). Specific cleavage can occur between Gln and Gly, with Gly-Pro remaining at the C terminus Ki-67-CR without any tag.

### Ki-67 consensus repeat expression and purification

The pDB-GFP-Ki-67-CR plasmid was transformed into *E. coli* BL21(DE3); transformed cells were grown overnight at 25 °C in N-5052 auto-induced medium[132], supplemented with 50 μg/ml kanamycin and 15 N NH4Cl. Cells were harvested by 20 min centrifugation at $6000 \times g$ at 4 °C. The pellet was resuspended in 20 mM Tris-HCl pH 7.5, 300 mM NaCl and 2 mM DTT (buffer A) and stored at −80 °C. Cells were supplemented with a Complete® EDTA free tablet (Roche), lysed by sonication, insoluble proteins and cell debris were sedimented by centrifugation at $40,000 \times g$ at 4 °C for 30 min. Supernatant was supplemented with imidazole to 5 mM final and loaded onto 5 ml gravity affinity columns (Ni Sepharose Excel 5 ml, Cytiva), equilibrated with buffer A. Columns were washed with 50 ml of buffer A and proteins were eluted with a one-step gradient of buffer B (buffer A containing 500 mM imidazole). The peak fractions were analysed by SDS-PAGE. Fractions containing tagged Ki-67-CR were pooled and dialysed overnight at 4 °C against buffer C (50 mM Bis-Tris pH 6.7, 50 mM NaCl, 2 mM DTT). The dialysed protein was then loaded on a Superdex S200 16/60 (HiLoad 16/600 Superdex 200 pg, Cytiva) equilibrated with buffer C. Fractions containing the protein of interest were analysed by SDS-PAGE and pooled. The purified GFP-Ki-67-CR protein was concentrated to 5 mg/ml with Vivaspin centrifuge concentrator (Sartorius Stedim Biotech).

### In vitro Ki-67 peptide phosphorylation assay

Protein was desalted by using PD10 Mini-Trap column in 50 mM Hepes 7.5, 50 mM NaCl, 2 mM DTT. Phosphorylation reaction was performed in a total volume of 220 μl, and contained 140 μl of GFP-Ki-67-CR (400 μg), 5 mM MgCl2, 500 μM ATP, 50 mM β-glycerophosphate, recombinant CDK1-cyclin B/CKS1 (5 μg; gift from Jane Endicott and Tony Ly), or recombinant GST-CDK2-cyclin A (ProQinase) in 50 mM Hepes pH 7.5, 50 mM NaCl, 2 mM DTT. Reaction was incubated at 30 °C for 18 h and stopped by adding 10 mM EDTA. It was subsequently desalted by using PD10 Mini-Trap column in 50 mM Bis-Tris pH 6.7, 50 mM NaCl, 2 mM DTT, and used to perform NMR experiments, Phostag SDS-PAGE and phosphoproteomics.

### NMR experiments and data analysis

All NMR samples contained final concentrations of 10% D2O and 0.5 mM 4,4-dimethyl-4-silapentane-1-sulfonic acid (DSS). Experiments were performed at 293 K on a Bruker Avance III spectrometer equipped with a cryogenic triple resonance probe and Z gradient coil, operating at a 1H frequency of 700 and 800 MHz. 15N-HSQC was acquired for each sample in order to determine amide (1HN and 15N) chemical shifts of non-phosphorylated and phosphorylated GFP-tagged Ki-67-CR. 15N-HSQC spectra were acquired for, respectively, non-phosphorylated (and phosphorylated) proteins at 800 (700) MHz using 32 (128) scans, 128 (256) increments and a spectral width of 22.5 ppm in the indirect dimension. All spectra were processed with Top-Spin v3.5 (Bruker Biospin) and analysed using CCPN-Analysis software[133]. Chemical shifts were referenced with respect to the H2O signal relative to DSS using the 1H/X frequency ratio of the zero point according to Markley et al.[134].

### Total internal reflection fluorescence (TIRF) experiments

In order to measure the ability of the phosphorylated and non-phosphorylated Ki-67 repeat to liquid–liquid phase separate, 50 μM protein was prepared in 50 mM Bis-Tris pH 6.7, 50 mM NaCl, 2 mM DTT. Dextran was added just before preparing samples on circular

glass coverslips (2.5 cm, 165-μm thick, Marienfeld). Coverslips were cleaned with a 15 min cycle of sonication with ultrasounds in 1 M KOH, followed by a second cycle of sonication in deionized water. Samples were deposited into wells of Press-to-seal silicone isolater with adhesive (Invitrogen), and covered with a second coverslip to avoid evaporation. Images were acquired with a custom-made TIRF microscope using a LX 488-50 OBIS laser source (Coherent). Oil immersion objective with a 1.4 numerical aperture (Plan-Apochromat 100×, Zeiss) was used. Fluorescence was collected with an EmCCD iXon Ultra897 (Andor) camera. The setup includes a 1.5× telescope to obtain a final imaging magnification of 150-fold, corresponding to a camera pixel size of 81.3 nm. Fluorescence images at different time points were obtained by averaging 150 individual images, each acquired over 50 ms exposure time.

### Phos-tag SDS-PAGE

For Phos-tag SDS-PAGE (12.5% Phos-tag™ SuperSep™ pre-cast gel 50 μmol/L; Fujifilm Wako Chemicals #193-16571), 500 ng of unphosphorylated and 500 ng of CDK1-cyclin B-CKS1 phosphorylated GFP-Ki-67-CR protein were loaded. SDS-PAGE was performed following a standard protocol, with the exception of two additional washes 20 min each in transfer buffer containing 10 mM EDTA, followed by a wash in transfer buffer, preceding wet transfer.

### Antibodies

The following commercial antibodies were used in this study:
MeCP2 (Abcam, ab253197; IF, 1:500)
H3K9me3 (Abcam, ab8868; IF, 1:500)
Nucleolin (Abcam, ab22758; IF, 1:1000)
Nucleophosmin (Abcam, ab183340; IF, 1:1000)
GFP (Chromotek, PABG1; Western blot, 1:10,000)
Alexa Fluor 568 conjugated goat anti-rabbit (Invitrogen, A11011; IF, 1:1000)
Goat anti-rabbit IgG (H + L) HRP (Thermo Fisher, # 32260; WB, 1:10,000).

### Mapping of Ki-67-CR phosphorylation sites by mass spectrometry

6.5 μg of GFP-Ki-67-CR protein phosphorylated by CDK1-cyclin B-CKS1 were digested in a FASP filter as described earlier for the other samples. Peptides were subsequently cleaned using C18 cartridges and phosphor-enriched using Fe(III)-NTA cartridges in the AssayMAP Bravo. Phosphopeptides were measured in a technical duplicate, acquiring in DDA mode with an Ultimate 3000 uHPLC system coupled to an Orbitrap Exploris 480 (Thermo Fisher Scientific) during a 60 min gradient. Raw files were searched against the Ki-67-CR sequence using MaxQuant with the same parameters as applied to the other phosphoproteomics experiments.

### Reporting summary

Further information on research design is available in the Nature Portfolio Reporting Summary linked to this article.

## Data availability

The mass spectrometry shotgun proteomics data generated in this study have been deposited in the ProteomeXchange Consortium via the PRIDE[135] partner repository with the dataset identifier PXD023310. The targeted proteomics data generated in this study have been deposited via Panorama[136] with the identifier PXD026088. The following public databases were used in this study: UniProt [https://www.uniprot.org/]; AlphaFold [https://alphafold.ebi.ac.uk/]; Phosphosite-Plus [https://www.phosphosite.org/homeAction.action]; Protein Data Bank [https://www.rcsb.org/]. Source data are provided with this paper.

## Code availability

Codes for rG-RPA calculation are uploaded in Github [https://github.com/MaxCalLab/IDPTheory/tree/main/rG-RPA]. Codes for phosphorylation and disorder analysis developed in this study are uploaded in Github [https://github.com/gero007/CDK-mediated-phosphorylation-of-IDRs].

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

## Acknowledgements

We thank Merlijn Witte for technical assistance with the *Xenopus laevis* egg fertilisation experiments, Ariane Abrieu for a gift of CSF egg extracts, Markus Raschle from the Technical University of Kaiserslautern for providing the *Xenopus laevis* protein database, Jonathan Huihui during early phases of theoretical analysis using SCDM, and Damien Coudreuse for critical reading of the manuscript. A.J.R.H. and M.A. acknowledge support from the Horizon 2020 program INFRAIA project Epic-XS (Project 823839) and the NWO funded Netherlands Proteomics Centre through the National Road Map for Large-scale Infrastructures program X-Omics (Project 184.034.019) of the Netherlands Proteomics Centre. J.M.V. is supported by scholarships from the Ministry of Science and Technology of Costa Rica (MICITT) and the University of Costa Rica (UCR). M.P. acknowledges support from NSF award number DMR-2213103. P.K. and M.V. are funded by the Oncode Institute which is financed by the Dutch Cancer Society and by the gravitation program CancerGenomiCs.nl from the Netherlands Organisation for Scientific Research (NWO). D.F. and L.K. are INSERM employees. GD is funded by the Institut National de Cancer, France (INCa) PRT-K programme (PRT-K17 n° 2018-023). The Fisher lab was funded by the Ligue Nationale Contre le Cancer, France (EL2018.LNCC/DF) and INCa (PLBIO18-094) and is currently supported by the Fondation ARC (N° ARCPGA12021010002850_3574). N.S. is supported by the ANR GPCteR (ANR-17- CE11-0022-01). D.F. and N.S. also acknowledge support from the I-SITE Excellence Program of the University of Montpellier, part of the Investissements France 2030. The CBS is a member of France-BioImaging (FBI) and the French Infrastructure for Integrated Structural Biology (FRISBI), supported by the French National Research Agency (ANR-10-INBS-04-01 and ANR-10-INBS-05). K.G. acknowledges support from NIH R01GM138901.

## Author contributions

M.A. and D.F. conceived and supervised the project. J.M.V., P.K. and L.K. designed and interpreted experiments. J.M.V., H.T., A.H., M.Ph., A.J.R.H., G.v.M., C.E.-R., A.F., N.A.S., E.A.G., D.C., M.Pa., A.B., N.S., L.K. and G.D. performed experiments and interpreted the data. M.V. and A.C. supervised G.v.M. and E.A.G., respectively. P.B. provided advice. J.M.V., L.K., G.D., K.G., D.F. and M.A. wrote the paper.

## Competing interests

The authors declare no competing interests.
