## [Peer Review File · Nature Communications]

REVIEWERS' COMMENTS

Reviewer #1 (Remarks to the Author):

The authors have submitted a revised version of their MS, which was originally submitted to Nature. In the revised version, they have made several changes that strengthen their narrative and their claims. They have clarified various aspects that were confusing in the original version. I read the responses and the revisions. Overall, the revisions seem to address the concerns raised by the reviewers. As far as my comments are concerned, I have nothing major to ask as revisions. There, however, are some aspects of Fig. 5 that are puzzling. It would help to address these in a final version.

The new aspects of the theory are interesting, but also puzzling. Please note that many of the proteins listed as IDPs are not substrates of CDKs. Further, the model assumes that these proteins are IDPs. Many feature IDRs, but they also feature ordered domains. NCL and NPM1 are two classic examples of this hybrid architecture. The theory does not make this distinction, which could be seen as a weakness.

As an aside, the responses by the authors were pugilistic and offensive. Everything that the reviewers wrote is was judged as being confused, incorrect, surprising, dismissive, or inaccurate. Yes, the review process is a frustrating one because we do not get our way. Instead of plaudits, we are asked questions, our assumptions and assertions are challenged, and clarity is sought. Every author and every manuscript goes through this level of scrutiny. I sympathize with the authors. They are disappointed. However, all four reviewers judged what was submitted, not what the authors wanted us to see. I have reviewed a lot, but this is honestly the first time I have felt that the process was not worth the effort.

Reviewer #1 (Remarks to the Author):

The authors have submitted a revised version of their MS, which was originally submitted to Nature. In the revised version, they have made several changes that strengthen their narrative and their claims. They have clarified various aspects that were confusing in the original version. I read the responses and the revisions. Overall, the revisions seem to address the concerns raised by the reviewers. As far as my comments are concerned, I have nothing major to ask as revisions.

We appreciate the reviewer's recognition that the paper has improved compared to the original and that it fully addresses concerns of all reviewers.

There, however, are some aspects of Fig. 5 that are puzzling. It would help to address these in a final version.

The new aspects of the theory are interesting, but also puzzling. Please note that many of the proteins listed as IDPs are not substrates of CDKs.

All of the human IDPs whose analysis was presented in Fig. 5 have previously been documented as CDK substrates, as explained in the text detailing our compilation of human CDK substrates based on the available literature (Suppl Data 2; see also Methods for sources). We only analysed the effect of phosphorylations that are confirmed as mediated by CDKs, as well as documented phosphorylations on the minimal CDK consensus motif. The exact phosphorylation sites are indicated.

Further, the model assumes that these proteins are IDPs. Many feature IDRs, but they also feature ordered domains. NCL and NPM1 are two classic examples of this hybrid architecture. The theory does not make this distinction, which could be seen as a weakness.

Our model does not assume that the proteins are entirely disordered, as explained in the text and in the figure legend. The term 'intrinsically disordered protein' does not exclusively denominate a protein that is 100% disordered; IDPs include both proteins that contain intrinsically disordered regions (IDRs), as well as entirely disordered proteins. Since we show that CDK phosphorylation is highly skewed to the disordered regions, which are thought to participate in protein-protein interactions, we only analysed the effects of phosphorylation on the disordered regions of the selected CDK targets. The analysed IDR sequences are indicated in the figure, and described in detail in Suppl. Data 5. We do, however, present the full-length Ki-67, where the size of the structured N-terminal FHA domain, which is not itself phosphorylated by CDKs, is negligible compared to the disordered remainder. To be clear, we do not make any claim about the effect of CDK-mediated phosphorylation on the structured regions within these proteins: it would indeed be useful to understand the effect of the phosphorylation on the entire protein entity, but no analytical theory exists yet that would allow modeling of the effect of phosphorylation of a single chain with an IDR and a folded domain.

As an aside, the responses by the authors were pugilistic and offensive.

We are very surprised that the reviewer perceived our replies in this manner, as they were not personal, but scientific.

Everything that the reviewers wrote is was judged as being confused, incorrect, surprising, dismissive, or inaccurate.

We feel that this comment is an exaggeration as in no way did we dismiss everything the reviewers wrote in our reply. It is perfectly normal to point out inaccuracies in the reviews, which are often a result of unclear explanation on our part in the version that was reviewed.

Yes, the review process is a frustrating one because we do not get our way.

It is not a question of getting one's way or not. In general, we do not find review processes frustrating, but when reviews use dismissive language (this particular reviewer used the phrase "Beyond Figure 1, the MS comes off the rails in terms of novelty", which no doubt reflects the reviewer's perception and which we did not take personally), it is surely understandable that we explain why we do not agree.

Instead of plaudits, we are asked questions, our assumptions and assertions are challenged, and clarity is sought. Every author and every manuscript goes through this level of scrutiny.

We have no problem with this whatsoever, and, no doubt like the reviewer, we have been through this process many times before, and understand it perfectly.

I sympathize with the authors. They are disappointed. However, all four reviewers judged what was submitted, not what the authors wanted us to see. I have reviewed a lot, but this is honestly the first time I have felt that the process was not worth the effort.

We are disappointed that the reviewer feels this way. We do not expect sympathy. We have also reviewed many papers, and indeed it is not an easy task to evaluate years of work condensed into a scientific paper in an insufficient space, especially when authors might be insufficiently clear. Yet one has to recognise that reviewers are not superhuman and cannot be experts on all aspects of a multidisciplinary study; they can also make mistakes; they may not have the time to read all of the relevant literature, and so on. We were of course disappointed with the editorial decision at *Nature* – who wouldn't be? – as we felt that we could revise the paper in accordance with the reviewers' criticisms, but in no way did we consider the reviews incompetent, and we apologise if the reviewer misunderstood our perception of his/her review.